

# Reflective properties of melt ponds on sea ice

Aleksey Malinka[1], Eleonora Zege[1], Larysa Istomina[2], Georg Heygster[2], Gunnar Spreen[2], Donald Perovich[3], and Chris Polashenski[4]

[1]Institute of Physics, National Academy of Sciences of Belarus, 220072, pr. Nezavisimosti 68-2, Minsk, Belarus
[2]Institute of Environmental Physics, University of Bremen, Otto-Hahn-Allee 1, D-28359 Bremen, Germany
[3]Thayer School of Engineering, Dartmouth College, Hanover, New Hampshire, USA
[4]Cold Regions Research and Engineering Laboratory, Engineer Research and Development Center, Hanover, New Hampshire, USA

*Correspondence to*: Aleksey Malinka (mal@light.basnet.by)

**Abstract.** Melt ponds occupy a large part of the Arctic sea ice in summer and affect strongly the radiative budget of the atmosphere-ice-ocean system. In this study the melt pond reflectance is considered in the framework of the radiative transfer theory. The melt pond is modeled as a plane-parallel layer of pure water upon a layer of sea ice (pond bottom). The reflection is considered as comprising of Fresnel reflection by the water surface and multiple reflections between the pond surface and its bottom, which is assumed to be Lambertian. Analytical formulas are put forward to calculate the bidirectional reflectance factor (BRF) and the albedo at different incident angles. The effects of the incident conditions and the atmosphere state are examined. The optical model developed is verified with data from *in situ* measurements made during several field campaigns performed on landfast and pack ice in the Arctic. The comparison to field spectra demonstrates good performance of the developed model for the variety of melt pond types observed in the Arctic.

## 1 Introduction

Melt ponds occupy a large fraction of the Arctic sea ice surface in summer (Barry, 1996; Perovich et al., 2009; Nicolaus et al., 2010; Polashenski et al., 2012). They reduce the ice albedo significantly and, therefore, increase the flux of absorbed sun light energy and speed up the process of melting, thus amplifying the positive ice-albedo feedback effect (Curry et al., 1995; Eicken et al., 2004; Pirazzini, 2008; Schröder et al., 2014). Including light reflection by melt ponds into climate models is an important task (Flocco et al., 2010; Flocco et al., 2012; Hunke et al., 2013; Lüpkes et al., 2013), particularly in times of the strong environmental changes we see nowadays (Serreze et al., 2000; Dethloff et al., 2006; Perovich et al., 2008; Pistone et al., 2014). A physical model of the reflective properties of melt ponds is needed for understanding the physics of sea ice, as well as for the correct interpretation of the results of remote sensing and field measurements (Herzfeld et al., 2006; Tschudi et al., 2008; Rösel et al., 2012; Warren, 2013; Zege et al., 2015).

Melt ponds on summer sea ice are also the most variable albedo-affecting factor: they can change from light blue ponds, when just formed, to dark mature ones (Perovich, 1996; Barry, 1996; Sankelo et al., 2010; Polashenski et al., 2012). Although there are quite a lot of measurements of melt pond spectral albedo (e.g., Perovich, 1994; Morassutti and Ledrew, 1996; Perovich et al., 2002, 2009), an adequate physical and optical model of melt pond reflection is still absent. Makshtas



and Podgorny (1996) gave the analytical formula expressing the pond albedo in terms of the albedo of its bottom. However, despite asserting that bottom albedo is the main factor that determines the albedo of a pond as a whole, they did not address how to calculate it. This essential gap exists up to now. In this work we propose the simple solution for the pond bottom spectral albedo. This solution has required the detailed consideration of the inherent optical properties of sea ice, which

forms the pond bottom. Beside, Makshtas and Podgorny (1996) give a formula for pond spectral albedo at direct incidence only; they do not consider the angular distribution of the reflected light. However, just the bi-directional reflectance is measured by satellite optical sensors. Besides, processing of the reflectance measurement data, both satellite and ground-based, requires the atmospheric correction, especially for polar regions. All these points are discussed in this work.

The paper is arranged as follows. Our model of melt pond reflectance is described in Sec. 2. Subsection 2.1 presents the

derivation of the formulas for pond reflectance, given by Makshtas and Podgorny (1996), expanded to various incident conditions. Inherent optical properties (IOPs) of sea ice are considered in subsection 2.2. Simple analytical solution for bottom albedo in terms of the ice IOPs and its thickness is given in subsection 2.3. Subsection 2.4 gives a final brief outline of the developed model. Accounting for the illumination conditions in processing and interpretation of the experimental results are considered in Sec. 3. The atmospheric correction of experimental data is considered in subsection 3.1. A

possibility to use the near IR reflectance as a banner of the ice grains presence is discussed in subsection 3.2. Notes about processing experimental data with common deficient information about incidence are given in subsection 3.3. Then, Sec. 4 presents the verification of the developed model with the three datasets of *in-situ* measurements (Polarstern-2012, Barrow-2008, and SHEBA-1998). The conclusion sums up the paper.

In this work we propose a simple optical model that enables the parameterization of the pond bottom albedo with a few

physical characteristics and thus determines the spectral reflective properties of the melt pond as a whole, including its bidirectional reflectance.

## 2   Model description

### 2.1     Radiance reflected by a melt pond

The model of reflection by melt ponds given in Makshtas and Podgorny (1996) uses the following assumptions:

25          1.   the water layer is an infinite plane-parallel layer;

2.   the melt water is pure, with neither absorbing contaminants nor scatterers;

3.   the Rayleigh scattering in water is negligible compared to the water absorption; a ray inside the pond is attenuated according to the exponential law;

4.   the pond bottom reflects light by the Lambert law (the reflected radiance is independent of the direction).

The described model is illustrated in Fig. 1. In this subsection we repeat the derivation of Makshtas and Podgorny (1996), expanding it to various illumination and observation conditions.

Let $E$ be the incident spectral irradiance. Then the light intensity (radiance) at the upper pond boundary is:

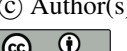



$$I_0 = \delta(\mu - \mu_0)\delta(\varphi)E,$$ (1)

where $\delta(x)$ is the Dirac δ-function, $\mu = \cos\theta$ ( $\mu_0 = \cos\theta_0$ ), $\theta$ is the observation zenith angle ( $\theta_0$ is the solar zenith

angle), $\varphi$ is the observation azimuth (counted from the solar principal plane).

The intensity of light reflected from the surface is:

$$I_0^R = R^F(\mu_0)\delta(\mu - \mu_0)\delta(\varphi)E,$$ (2)

where $R^F(\mu_0)$ is the Fresnel reflectance for incidence angle $\theta_0$.

When the ray of intensity $I_0$ at incident angle $\theta_0$ is refracted by the pond boundary, the angle of refraction $\theta_0^w$ is given by

Snell's law:

$$\theta_0^w = \arcsin\left(\frac{1}{n}\sin\theta_0\right),$$
$$\mu_0^w = \frac{1}{n}\sqrt{n^2 - 1 + \mu_0^2},$$ (3)

and the intensity of light just under the boundary is given by the law of conservation of basic radiance:

$$I_0^{w\downarrow} = T^F(\mu_0)I_0 n^2,$$ (4)

where $T^F(\mu_0)$ is the Fresnel transmittance for incidence angle $\theta_0$.

The light field inside the pond can be divided into the up- and down-welling fluxes. The up-welling flux is the light reflected

by the pond bottom. The intensity of the down-welling light inside the pond $I^{w\downarrow}$ consists of the direct light intensity $I_0^{w\downarrow}$

(direct solar light, refracted by the pond surface) and the diffuse light intensity $I_1^{w\downarrow}$ (the light that was subjected to

reflections between the bottom WI and the surface AW):

$$I^{w\downarrow} = I_0^{w\downarrow} + I_1^{w\downarrow}.$$ (5)

The direct light flux, incident to the pond bottom, is:

$$F_0^w = T^F(\mu_0)E\mu_0\exp\left(-\frac{\varepsilon_w z}{\mu_0^w}\right),$$ (6)

where $\varepsilon_w$ is the extinction coefficient of water, equal to the sum of the water absorption ( $\alpha_w$ ) and scattering ( $\sigma_w$ )

coefficients:

$$\varepsilon = \alpha_w + \sigma_w.$$ (7)

We use the data of Segelstein (1981) for the water absorption and the power law for the spectral scattering coefficient:

$$\sigma_w(\lambda) = \sigma_0\left(\frac{\lambda_0}{\lambda}\right)^{4.3},$$ (8)

where $\lambda$ is the wavelength and $\sigma_0 = 1.7 \times 10^{-3} m^{-1}$, $\lambda_0 = 550 nm$ (Kopelevich, 1983).





The intensity of up-welling light just below the AW interface is:

$$I_1^{w\uparrow}(\mu^w,0) = I_1^{w\uparrow}(\mu^w,z)\exp\left(-\frac{\varepsilon_w z}{\mu^w}\right).$$

(9)

After the internal reflection the intensity of the down-welling light just below the AW interface is:

$$I_1^{w\downarrow}(\mu^w,0) = R_{in}(\mu^w)I_1^{w\uparrow}(\mu^w,0) = R_{in}(\mu^w)I_1^{w\uparrow}(\mu^w,z)\exp\left(-\frac{\varepsilon_w z}{\mu_0^w}\right),$$

(10)

where $R_{in}$ is the internal reflection coefficient.

The intensity of the down-welling diffuse light at the pond bottom is

$$I_1^{w\downarrow}(\mu^w,z) = R_{in}(\mu^w)I_1^{w\uparrow}(\mu^w,z)\exp\left(-2\frac{\varepsilon_w z}{\mu_0^w}\right).$$

(11)

As the bottom is Lambertian, the intensity of the up-welling light just above the bottom is independent of direction:

$$I_1^{w\uparrow}(\mu^w,z) = I_1^{w\uparrow}(z).$$

(12)

The boundary conditions at the pond bottom are:

$$\pi I_1^{w\uparrow}(z) = \pi A_b f_{in}(\varepsilon_w z)I_1^{w\uparrow}(z) + A_b F_0^w,$$

(13)

where $A_b$ is the bottom albedo and

$$f_{in}(x) = 2\int_0^1 R_{in}(\mu^w)\exp\left(-2\frac{x}{\mu^w}\right)\mu^w d\mu^w.$$

(14)

From Eq. (13) we have:

$$I_1^{w\uparrow}(z) = \frac{A_b F_0^w}{\pi\left(1 - A_b f_{in}(\varepsilon_w z)\right)}.$$

(15)

Putting together Eqs. (4)-(15), we get for the intensity of light $I_1^R$ that goes out from the pond:

$$I_1^R(\mu) = \frac{E\mu_0 T^F(\mu)T^F(\mu_0)A_b}{\pi n^2\left(1 - A_b f_{in}(\varepsilon_w z)\right)}\exp\left(-\frac{\varepsilon_w z}{\mu_0^w} - \frac{\varepsilon_w z}{\mu^w}\right).$$

(16)

The total intensity of light reflected by the melt pond is:

$$I^R = I_0^R + I_1^R = R^F(\mu_0)\delta(\mu-\mu_0)\delta(\varphi)E + \frac{E\mu_0 T^F(\mu)T^F(\mu_0)A_b}{\pi n^2\left(1 - A_b f_{in}(\varepsilon_w z)\right)}\exp\left(-\frac{\varepsilon_w z}{\mu_0^w} - \frac{\varepsilon_w z}{\mu^w}\right).$$

(17)

The bidirectional reflectance factor (BRF) by definition is equal to:

$$R = \frac{\pi I^R}{\mu_0 E}.$$

(18)

Hence the BRF of a melt pond is:

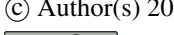


$$R = \frac{\pi}{\mu_0} R^F(\mu_0)\delta(\mu - \mu_0)\delta(\varphi) + \frac{T^F(\mu)T^F(\mu_0)A_b}{n^2\left(1 - A_b f_{in}(\varepsilon_w z)\right)}\exp\left(-\frac{\varepsilon_w z}{\mu_0^w} - \frac{\varepsilon_w z}{\mu^w}\right). \tag{19}$$

The first term describes the sun glint from the AW surface; the second one describes the light, multiply reflected between the pond bottom WI and the surface AW.

The albedo at direct incidence $A(\mu_0)$ (the black-sky albedo)

$$A(\mu_0) = \frac{1}{\pi}\int_0^{2\pi}\int_0^1 R(\mu)\mu\,d\mu\,d\varphi. \tag{20}$$

is found by means of integrating Eq. (19):

$$A(\mu_0) = R^F(\mu_0) + \frac{T^F(\mu_0)f_{out}(\varepsilon_w z)A_b}{n^2\left(1 - A_b f_{in}(\varepsilon_w z)\right)}\exp\left(-\frac{\varepsilon_w z}{\mu_0^w}\right), \tag{21}$$

where

$$f_{out}(x) = 2\int_0^1 T^F(\mu)\exp\left(-\frac{x}{\mu^w}\right)\mu\,d\mu. \tag{22}$$

The albedo at diffuse incidence $A^D$ (the white-sky albedo)

$$A^D = 2\int_0^1 A(\mu_0)\mu_0\,d\mu_0. \tag{23}$$

is found by integrating Eq. (21):

$$A^D = R^{FD} + \frac{f_{out}^2(\varepsilon_w z)A_b}{n^2\left(1 - A_b f_{in}(\varepsilon_w z)\right)}, \tag{24}$$

where $R^{FD}$ is the Fresnel reflectance for the diffuse incidence.

Functions $f_{in}(x)$ and $f_{out}(x)$ are related by the equation:

$$f_{out}(2x) = n^2\left(2E_3(2x) - f_{in}(x)\right), \tag{25}$$

where $E_3(x)$ is the integral exponential function of the third power:

$$E_3(x) = \int_1^\infty \frac{e^{-xt}}{t^3}\,dt. \tag{26}$$

However, the relationship (25) is not very useful in numerical calculations, because these functions are evaluated at different argument values ($x$ and $2x$).



## 2.2    Inherent optical properties of substrate ice

The main factor in Eqs. (19), (21), and (24) that determines the melt pond reflection is its bottom albedo $A_b$. In order to calculate it we should first consider the inherent optical properties (IOPs) of substrate ice that forms the pond bottom.

The IOPs of a medium used in the radiative transfer theory are the spectral scattering $\sigma(\lambda)$ and absorption $\alpha(\lambda)$

coefficients and the scattering phase function $p(\theta)$. In the following consideration, as in other numerous radiative transfer theory applications, the transport scattering coefficient $\sigma_t$ is used:

$$\sigma_t = \sigma(1-g),\tag{27}$$

where $g$ is the average cosine of the scattering angle $\theta$:

$$g = \langle \cos\theta \rangle = \frac{1}{2}\int_0^\pi p(\theta)\cos\theta\sin\theta\,d\theta.\tag{28}$$

The transport coefficient is useful in calculating the reflection and transmission by a scattering layer with very elongated phase function, particularly if one is interested in the layer albedo, rather than the angular structure (BRF) of the reflected light (Zege et al., 1991).

Main factors that determine optical properties of sea ice are its microphysical structure and values of complex refractive indices of its constituents; the dispersion of complex refractive indices determines the spectral properties of sea ice.

As the volume concentration of air bubbles in sea ice is small – only up to ~5% even in the extremely bubbly ice (Gavrilo and Gaitskhoki, 1970) – and the complex refractive index of brine is very close to that of ice (see Buiteveld et al., 1994; Warren and Brandt, 2008; and Sec. 2.2b), we take the absorption coefficient of sea ice equal to that of solid ice. Impurities – sediment and organic pigments from sea water – could change absorption coefficients, particularly at shorter wavelengths. At this stage we neglect their effect, keeping in mind that their absorption spectra can be easily added, if necessary.

The scattering takes place at inhomogeneities in sea ice and is mainly caused by air bubbles and brine inclusions (Mobley et al., 1998; Light, 2010). Another source of scattering could be salt crystals, but they precipitate at low temperatures and are not observed in summer ice, where melt ponds are formed: mirabilites ($Na_2SO_4\ 10H_2O$) begin to precipitate at $-8^0C$ and hydrohalites ($NaCl\ 2H_2O$) at about $-23^0C$ (Light et al., 2003).

*a) Air bubbles*

The upper layer of sea ice (20-30 cm) contents usually significant amount of air bubbles (Gavrilo and Gaitskhoki, 1970; Mobley et al., 1998), with volume concentration, which can reach values of 5% and decreases with depth. (We do not consider here the highly scattering surface layer that forms in the melting process and is commonly referred to as 'white ice'). Air bubbles in sea ice are mostly spherical. Light (2010) gives the following size distribution for bubbles in the first year ice:



$$N(r) \sim r^{-1.5}, \qquad 4\mu m \leq r \leq 70\mu m. \tag{29}$$

Gavrilo and Gaitskhoki (1970) report the presence of much larger bubbles in the bubbly ice (from 0.1 to 2 mm with the exponent 1.24).

However, since air bubbles in ice are optically hard (the refractive index of air differs strongly from that of ice) and do not

absorb light, scattering by bubbles of this size range is described by the laws of geometrical optics. Thus, the scattering characteristics do not depend on the bubble size (unless considering the strictly forward and backward directions), the shape of the size distribution is also insignificant. Particularly, the scattering efficiency $Q_{sca}$ in this limit equals 2 and the phase function can be easily calculated with the Mie formulas for any type of size distribution, e.g., for the one given in Eq. (29). The refractive index of air (relative to ice) in the interval 0.35-0.95 μm changes from 0.755 to 0.768 with average value of

0.763 within this interval. The corresponding average cosine $g$ , obtained with the Mie calculations, takes values from 0.851 to 0.865 with the mean value of 0.860, and therefore the spectral variability does not exceed 2%.

*b) Brine inclusions*

The main features of brine inclusions are the following: they are optically soft, i.e., their refractive index $n_b$ (brine relative to ice) is close to unity; their size is large (comparing to the wavelength); and their shape is strongly irregular. Implying these

features we can apply the approximation for irregularly shaped particles developed by Malinka (2015) to describe scattering properties of brine inclusions.

The size of brine inclusions, which can be of the order of several millimeters, is so much larger than the wavelength of visible light that their optical properties can be considered in the limit of infinitely large particles, despite their refractive index $n_b$ is close to unity:

$$n_b - 1 \ll 1. \tag{30}$$

In this limit the size distribution is also insignificant and the scattering efficiency $Q_{sca}$ is independent of the wavelength:

$$Q_{sca} = 2. \tag{31}$$

The scattering phase function of optically soft particles can be approximated according to Malinka (2015) by:

$$p(\theta) = \frac{2x^2(1+\mu^2)}{\left(1+2x^2(1-\mu)\right)^2}, \tag{32}$$

where $\mu = \cos\theta$ and $x$ is the optical particle size (dimensionless), which for large particles equals:

$$x = \frac{n_b+1}{n_b-1}\sqrt{\frac{Q_{sca}}{8}}. \tag{33}$$

In view of Eqs. (30) and (31), we can write:



$$x = \frac{1}{n_b - 1}. \tag{34}$$

The average cosine $g$ of the phase function (32) is:

$$1 - g = \frac{\log 2x - 1}{x^2}. \tag{35}$$

Figure 2 demonstrates the spectral dependence of the refractive index of water relative to ice. We used the data by Warren and Brandt (2008) for ice. The refractive index of water is taken from Hale and Querry (1973), Segelstein (1981), Daimon and Masumura (2007), and Kedenburg et al. (2012) for distilled water, formula of Quan and Fry (1995) for brine of zero salinity at temperature $0^0 C$, and formula of Frisvad (2009), which is based mainly on Quan and Fry (1995) and the measurements by Maykut and Light (1995), for brine with equilibrium salinity at temperatures $-2^0 C$, $-4^0 C$, and $-6^0 C$. The earlier data (Hale and Querry, 1973 and Segelstein, 1981) clearly demonstrates the spectral dependences, their dispersions being opposite. In contrast, the newer data do not demonstrate such dependence: according to the more modern data the refractive index (relative to ice) of water, including brine, is almost spectrally neutral. This question is important when describing the light scattering by brine inclusions in ice, because the transport scattering coefficient is determined mainly by the value $(n_b - 1)^2$ (see Eqs. (27) and (34)-(35)). Finally, according to the newer data we will accept that the relative refractive index of brine, and therefore the transport scattering coefficient of brine inclusions, is spectrally neutral. E.g., $n = 1.024$ for temperature $-2^0 C$ and, according to Eqs. (34)-(35), $g = 0.998$. Note that the value of $g$ in this model is significantly greater than that used in many other studies, e.g., in Mobley et al., (1998) or Light et al., (1998).

*c) Inherent optical properties of sea ice*

Light scattering properties of sea ice are a combination of those of brine inclusions and air bubbles. The total and transport scattering coefficients are the sum of the respective values:

$$\sigma = \sigma_b + \sigma_a,$$
$$\sigma_t = \sigma_b^t + \sigma_a^t. \tag{36}$$

We denote the values related to brine inclusions with the subscript $b$ and to air bubbles with the subscript $a$. The phase function and the average cosine are the linear combination of the respective values:

$$p(\theta) = \frac{\sigma_b}{\sigma} p_b(\theta) + \frac{\sigma_a}{\sigma} p_a(\theta),$$
$$1 - g = \frac{\sigma_b}{\sigma}(1 - g_b) + \frac{\sigma_a}{\sigma}(1 - g_a) = \frac{\sigma_t}{\sigma}. \tag{37}$$

Once $g_a$ and $g_b$ are known (e.g., at $-2^0 C$ $g_a = 0.86$, $g_b = 0.998$), the resulting $g$ depends only on the proportion of fractions $a$ and $b$.



Generally, the IOPs of sea ice depend on its microstructure. In view of the fact that both bubble and brine inclusion size is much larger than the wavelength, the scattering coefficient equals:

$$\sigma_j = 2\psi_j \qquad (j=a,b),$$ (38)

where $\psi$ is the specific cross-sectional area of inclusions (air or brine):

$$\psi_j = \langle S_\perp \rangle_j N_j = \frac{3 C_j^V}{4 R_j}.$$ (39)

Here subscript $j$ shows the fraction number, $\langle S_\perp \rangle_j$ is the average cross-sectional area of $j$-inclusions, $R_j$ their effective radius, $N_j$ and $C_j^V$ are their numeric and volume concentration, respectively.

The phase function (and consequently its average cosine $g$) can be characterized by the ratio of volume concentration air-to-brine $C_a^V / C_b^V$, if their effective radii are determined. E.g., for bubbles size distribution (29), the effective radius is $R_a = 42.55 \mu m$. Light (2010) gives the value of 110 m$^{-1}$ for $\psi_b$, the specific cross-sectional area of brine inclusions, for a sample of typical first year ice at $-15^0$C, which can grow up to 400 m$^{-1}$ when warming. The estimate, made by Light (2010) for the brine volume concentration in the same sample, gives the values from 1.2% to 1.9%. This allows us to estimate the effective radius of brine inclusions as $R_b \approx 100 \mu m$.

However, as the morphology of sea ice can vary drastically with place and time, the more convenient way to characterize the ratio of air and brine fractions is to use the ratio of their transport coefficients $\sigma_a^t / \sigma_b^t$. This ratio is related to the ratio of volume concentrations as:

$$\frac{\sigma_a^t}{\sigma_b^t} = \frac{1-g_a}{1-g_b} \frac{R_b}{R_a} \frac{C_a^V}{C_b^V}.$$ (40)

Figure 3 presents the phase function of mixtures with different air-to-brine fractions ratio.

We conclude that the phase function (and consequently $g$) of sea ice is spectrally neutral in the visible and near IR range. In virtue of Eq. (38), the scattering coefficient $\sigma$ is also spectrally neutral. Consequently, the transport scattering coefficient $\sigma_t$ is also spectrally neutral and can serve as a scalar parameter that characterizes scattering in sea ice.

## 2.3 Bottom albedo

If both the absorption and transport scattering coefficients are known, the albedo of a layer can be calculated within the two-stream approximation, which is widely used for practical calculations:

$$A_b = A_0 \frac{1-\exp(-2\gamma\tau)}{1-A_0^2 \exp(-2\gamma\tau)},$$ (41)



where $A_0$ is the albedo of the semi-infinite layer with the same optical characteristics, $\gamma$ is the asymptotic attenuation coefficient, and $\tau$ is the layer optical thickness. The version of the two-stream approximation developed by Zege et al. (1991) expresses these characteristics as follows:

$$A_0 = 1 + t - \sqrt{t(t+2)},$$
$$\gamma = \frac{3}{4}\frac{\sigma_t}{\sigma_t + \alpha_i}\sqrt{t(t+2)}, \tag{42}$$
$$\tau = (\sigma_t + \alpha_i)H,$$

with
$$t = \frac{8\alpha_i}{3\sigma_t}, \tag{43}$$

where $\alpha_i$ is the ice absorption coefficient; $H$ is the ice layer thickness.

The two-stream approximation in the version given in Zege et al. (1991) has a wide range of applicability and can be used both for strongly and weakly absorbing media, for optically thin and thick layers. Hence, this approximation can be applied to all the variety of melt ponds: from young ponds, which are light blue and have comparatively optically thick ice substrate,

to mature dark ones, where substrates are optically thin.

### 2.4    Model outline

Thus, in the assumption of a Lambertian bottom and plane parallel geometry, which assumes the absence of strong wind, i.e., calm pond surface, the spectral reflection of ponds is determined by two values: water layer depth $z$ and the albedo of the pond bottom $A_b$. The latter, in turn, depends on the transport scattering coefficient of substrate ice $\sigma_t$ and its geometric

thickness $H$ (or, respectively, the transport optical thickness $\sigma_t H$). Note that only value $\alpha_i$ in Eqs. (41)-(43) has a spectral behavior, while the others – $\sigma_t$ and $H$ – are scalars.

The outlined model of a melt pond is shown in short in Table 1.

## 3    Illumination conditions

### 3.1    Atmospheric correction

Correct processing of the reflection measurement results requires the correct modeling of the illumination conditions. This is especially important for measurements in the Arctic, because of the low sun and the bright surface. When the sky is overcast, the incident light is close to diffuse, even if the solar disk is visually observed (Malinka et al., 2016b). In this case the measured albedo is the white-sky one. However, when the sky is clear and the sun is near the horizon, the direct solar flux is comparable to the diffuse flux from the sky, so the measured (blue-sky) albedo value is a mixture of those at direct (black-

sky) and diffuse (white-sky) incidence. The black-sky albedo increases when the sun is approaching the horizon, so the





difference between the white- and black-sky albedos is most essential at oblique incidence (see Fig. 4). The problem of the correct interpretation of the measured blue-sky albedo is considered in detail in Malinka et al. (2016b) for a homogeneous surface. However, the albedo of a melt pond can differ significantly from that of the surrounding background, e.g., white ice or snow. Some estimation for this case is given below.

Let $R$, $A(\mu_0)$ and $A^D$ be, as before, the BRF, black-sky, and white-sky albedo of a melt pond, respectively. Let the surrounding background be Lambertian with albedo $r_b$. Then the brightness of the incident radiance can be estimated as (Malinka et al., 2016b):

$$B_\downarrow = \left[ t_0(\mu_0)\hat{\delta} + t_d(\mu_0) + \frac{r_a r_b}{1 - r_a r_b} T(\mu_0) \right] \frac{E_0 \mu_0}{\pi},$$  (44)

where $t_0(\mu_0)$ and $t_d(\mu_0)$ are the direct and diffuse atmosphere transmittances, $\hat{\delta} = \pi \delta(\mu - \mu_0)\delta(\varphi - \varphi_0)/\mu_0$ is the identity

operator ($\delta(x)$ is the Dirac delta-function), $T(\mu_0) = t_0(\mu_0) + t_d(\mu_0)$ is the atmosphere transmittance at direct incidence, and $r_a$ is the atmosphere bihemispherical reflectance at incidence from below. $E_0$ is the extra-terrestrial solar irradiance.

So, the light flux incident to a melt pond is:

$$F_\downarrow = \frac{T(\mu_0)}{1 - r_a r_b} E_0 \mu_0.$$  (45)

The radiance of light reflected by pond follows from Eq. (44):

$$B_\uparrow = \left[ R(\mu, \mu_0, \varphi) t_0(\mu_0) + A(\mu)\left( t_d(\mu_0) + \frac{r_a r_b}{1 - r_a r_b} T(\mu_0) \right) \right] \frac{E_0 \mu_0}{\pi}$$

(46)

$$= \left[ \left( R(\mu, \mu_0, \varphi) - A(\mu) \right) t_0(\mu_0) + A(\mu) \frac{T(\mu_0)}{1 - r_a r_b} \right] \frac{E_0 \mu_0}{\pi}.$$

Therefore the reflected flux is:

$$F_\uparrow = \left[ \left( A(\mu_0) - A^D \right) t_0(\mu_0) + A^D \frac{T(\mu_0)}{1 - r_a r_b} \right] E_0 \mu_0.$$  (47)

For the measured value of the blue-sky albedo $\alpha$ it follows:

$$\alpha = \frac{F_\uparrow}{F_\downarrow} = \frac{\left( A(\mu_0) - A^D \right) t_0(\mu_0)(1 - r_a r_b) + A^D}{T(\mu_0)}.$$  (48)

The equation for the blue-sky albedo can be written as a linear combination of the black and white-sky albedos:

$$\alpha = w A(\mu_0) + (1 - w) A^D,$$  (49)

with the proportion of direct radiance $w$:

$$w = \frac{t_0(\mu_0)}{T(\mu_0)}(1 - r_a r_b).$$  (50)



Factor $(1 - r_a r_b)$ is responsible for multiple reflections between the atmosphere and surrounding background.

Albedo spectra of a light melt pond (a pond with high reflectance) at different illumination conditions are shown in Fig. 5. The angle of incidence is $80^0$ (the sun elevation is $10^0$). The interval of albedo changes is limited by the values of white and black-sky ones. Also shown are the blue-sky albedos for clear sky and for sky with thin cirrus layer (with optical thickness of

0.1). Both are considered with different surrounding backgrounds: perfectly black ($r_b = 0$) and white ($r_b = 1$). As seen from Fig. 5, the effect of background is negligible, so the results of melt pond albedo measurements can be processed without *a priori* knowledge of the albedo of surrounding background.

## 3.2    IR reflectance

In contrast to the visible range, ice and water absorb a significant amount of light in the IR: a few centimeter thick layer of

ice or water completely absorbs radiation in the infrared range. Thus the melt pond optical response in the IR is restricted to the Fresnel reflection by the pond surface. In contrast, ice grains in white ice are of the order of millimeters in size (and even smaller in snow). Due to this fact one can trace the appearance of the specific features of the behavior of the imaginary part of the ice refractive index $\kappa$ in the IR in the reflection spectra of white ice and snow. In particular, $\kappa$ has a local minimum at 1.1 μm, which provides a slight peak of reflection in the interval 1.05-1.11 μm (Wiscombe and Warren, 1980). Figure 6

shows an example of the albedo spectral dependence for white ice, snow, and a melt pond. It clearly demonstrates that for wavelengths longer than 0.9 μm the melt pond reflection is restricted by the Fresnel reflection to a constant value, while snow and white ice demonstrate a local maximum at 1.1 μm. Thus, this slight peak can serve as a criterion for determining if a spectrum is taken entirely from an open pond or partially from snow/ice surface. If this peak is observed in a measured spectrum, it clearly indicates the presence of ice grains (of white ice or snow) in the receiver field of view.

## 3.3    Measurement geometry

In the description of the field data used in this study most sky conditions were reported as overcast. Only a few measurements were taken under clear sky conditions. Scattered clouds were not reported at all in the measurement series considered, likely due to challenges collecting accurate albedo measurements in variable illumination. In the cases of overcast sky, the measured albedo was interpreted as the white-sky one. In the clear sky cases, the Rayleigh atmosphere with

the Arctic Background aerosol (Tomasi et al, 2007) was assumed. In this case the incident angle was determined from the pond reflection in the IR: at the interval 1.25-1.3 μm (preferably) or 0.85-0.9 μm, if data at the former interval are not available. As the IR signal (both incident and reflected) is quite weak and hence some noise is always noticeable, we average the signal over one of the abovementioned intervals. The pond reflectance in these IR intervals is completely determined by the Fresnel reflection of its upper boundary. Atmosphere scattering in the IR is negligible (especially at 1.3 μm), so the

incident light is unidirectional. In this situation the incident angle can be easily calculated through the Fresnel formulas.



## 4    Model verification

Three different datasets with in-situ field measurements were used for the evaluation of the pond model. They are described in the next subsections.

### 4.1    Polarstern-2012

The measurements of spectral albedo of the Arctic surfaces were carried out during the R/V *Polarstern* cruise ARK-XXVII/3 (August 2 – October 8, 2012). Only in the second half of the cruise did the vessel leave the marginal ice zone and enter the ice pack. The ice thickness varied from 0.5 to 3 meters with an average of 1-1.5 meter. The melt ponds observed were both open and frozen over, sometimes snow covered. The data were collected during stations, when the vessel was parked at an ice floe for several days. This gave the possibility to obtain several-day data sequences of melting sea ice and

forming melt ponds at the same location.

The ASD FieldspecPro III spectroradiometer used for these measurements has three different sensors that provide measurements from 350 nm to 2500 nm with the spectral resolution of 1.0 nm. A 10x10 cm$^2$ *Spectralon* white plate served as a diffuser, which was held at about 1 meter above the surface and was directed first towards the measured surface and then towards the sky. The ratio of these two measurements gives the hemispherical reflectance (albedo) of the surface. For some

cases the water depth and ice thickness within the pond were measured.

For the model verification we considered the melt pond albedo in the spectral interval 0.35 – 1.3 μm. The retrieval procedure implies searching for the pond parameters values (see Table 1) that provide the best fit (in the sense of the least squares) of the measured and modeled spectra. For the cases where the pond depth and underlying ice thickness were known the retrieved pond parameters were compared to the measured ones.

Some ponds were frozen over, i.e., they had a layer of newly formed ice on top of their surface. It is evident that a layer of transparent ice at the pond surface practically does not change pond reflection, so we consider the ponds with ice crust in the same manner as open ones. However, if the upper ice layer is bubbly or snow covered, the pond reflectance can change drastically: the pond gets brighter and may become indistinguishable from the surrounding ice in the visible range. These snow-covered ponds would require other means for their characterization. We exclude such cases from the consideration.

Figures 7-10 present photos of different ponds and their reflectance spectra, measured and simulated with the retrieved parameters (denoted as 'retrieved' in the legend).

Figure 7 shows the photos, modeled and measured spectra of light blue melt ponds with uniform bottom on thick first-year ice under clear and cloudy skies, measured in the Central Arctic on 10.08.2012, 10.08.2012 and 22.08.2012, respectively. In all cases the ponds are frozen over with a 2-3 cm layer of ice on top. Figure 8 shows three cases of frozen over blue ponds

with heterogeneous bottom under overcast skies measured on 11.08.12, 22.08.12 and 22.08.12, respectively. One can see darker parts in the ponds, which result from sea ice melting from the lower boundary. Figure 9 presents dark open melt ponds on thinner first year ice under overcast skies, all measured on 26.08.2012. The albedo of these ponds is much lower



than that of the previous ones: from about 0.07 to 0.14 in the visible and about 0.05 in the IR. Figure 10 presents the two cases of light blue ponds both measured on 26.08.12 and a dark pond contaminated with algae aggregates measured on 21.08.2012, all under overcast skies. Surprisingly, the spectrum of the pond with algae is reproduced quite well. This is because the contribution of the yellow algae spots to a total reflection is proportional to their area, which is not very large.

However, their effect can be clearly seen in the spectrum: the measured values are less than the modeled ones in the blue range (0.3-0.5 μm) and greater in the yellow-green (0.5-0.6 μm).

The above ponds are quite different: from dark to very light blue in color, open and frozen over, clear and contaminated with organic matter. In spite of this, the model is able to reproduce the measured spectra in the visible region with high accuracy in all studied cases. The root-mean-square difference (RMSD) between the measured and simulated spectra has the average

value of 0.01 for the whole considered spectrum and 0.007 for the visible range.

The retrieved and measured geometrical parameters of the ponds, as well as the RMSD between the measured and simulated spectra, are presented in Table 2 and shown in Fig. 14.

## 4.2    Barrow-2008

Melt pond spectra observed in Barrow were collected as part of the SIZONET program observing pond formation

(Polashenski et al., 2012). Observations were collected at sites approximately 1 km offshore from Niksiuraq on landfast ice in the Chukchi sea, near 71.366N, 156.542W on level first year ice. For this work, a total of 27 measured melt pond spectra were used (no photographs were taken). All melt ponds were quite dark and polluted with sediments and their spectra look quite similar. Three of them are presented in Fig. 11. The albedo does not exceed the value of 0.3 in its maximum and show a discrepancy in the blue range, presumably due to the presence of mineral sediments. Because of this, the RMSD between

the measured and simulated spectra for the visible range (0.01) is greater than that for the whole spectrum (0.009). The ice thickness was not measured. The pond depths, measure and retrieved, as well as the RMSD, are shown in Table 2 and Fig. 14.

## 4.3    SHEBA-1998

SHEBA was a year-long drift experiment conducted in the Beaufort Sea from October 1997 – 1998 (Perovich et al., 1999;

Uttal et al., 2002). Extensive measurements of the properties and processes of the atmosphere-ice-ocean system were made. This included observations of the spatial variability and temporal evolution of the spectral albedo of the ice cover (Perovich et al., 2002).

One pond in this expedition was especially interesting, because its bottom had a region that was much brighter than the surrounding bottom. This region had sharp borders and rectangular corners (see the photo in Fig. 12). This likely was a

broken piece of bubbly multiyear ice that was incorporated into the ice cover. This piece of ice had more air bubbles than the darker adjacent ice. This dual pond was observed during the entire period of its formation and development; the spectra were





taken every four days. The most intensive formation process was observed from July 17 through August 14. The spectra taken during this period were processed and the results are shown in Figs. 12 and 13.

Figure 12 shows the spectra and the photos of the SHEBA dual pond. For the first five dates (July 17, 21, 25, 29, August 2) the retrieval is excellent (for the visible range RMSD = 0.0038 for July 17 and has a maximal value of 0.0061 for July 29, see Table 2) and for the last three (August 6, 10, 14) the retrieval is a little bit worse, but still quite good (for the visible range RMSD = 0.0085 for August 6 and 10). The reason for this difference is not obvious and we may assume that some contaminant got into the pond those days. So, the regression analysis relies on the first five measurement dates.

Figure 13 presents the retrieved pond depth and ice thickness (for both parts independently) for these dates. The retrieved pond depth at the light part is 7 cm greater and at the dark one is 13 cm greater than the average reported pond depth (37 cm). Albedo of the light part (in the visible part of spectrum) is approximately twice greater than that of the dark part. In general, this agrees with the different nature of their physical properties. The retrieved ice thickness in the light part is lower by 34 cm in average than that of the dark part. The slope of the linear regression for the retrieved ice thickness gives the melt rate of 1.9 cm/day and 2.6 cm/day for the light and dark parts, respectively. Taking the average surface and bottom melt for SHEBA ponded ice from 17 July to 14 August gives an estimated surface ice melt of 35 cm and bottom melt of 28 cm for a total of 63 cm, which gives a melt rate of 2.25 cm/day (Perovich et al., 2003).

Suppose that the difference between the transport scattering coefficient $\sigma_t$ for the light and dark portion is due to air bubbles only, then the scattering coefficient by air bubbles can be calculated as:

$$\sigma_a = \frac{\sigma_t^{light} - \sigma_t^{dark}}{1 - g_a} \, . \tag{51}$$

For the first five dates the average retrieved scattering coefficient by air bubbles is 33 m$^{-1}$, the slope being much less than the scatter. In the bubble saturated ice observed by Gavrilo and Gaitskhoki (1970) the air volume concentration was up to 5% and the effective bubble radius was $R_a = 1.3$ mm. If we suppose the same effective radius, the average air volume concentration in the light ice will be $C_a^v = 2/3 R_a \sigma_a = 2.8\%$, which is quite reasonable for bubbly ice.

## 4.4    Verification results

The retrieved and measured pond parameters (melt water depth and underlying ice thickness), as well as root mean square difference (RMSD) between the measured and simulated albedo spectra, are given in Table 2. The RMSD is shown both for the whole spectrum and for the visible range ($\lambda < 0.73 \mu m$). The scatter plot of the retrieved pond parameters is shown in Fig. 14. The retrieval of the underlying ice thickness is made with reasonable accuracy; the maximal error is 55%, the relative RMSD is 37% and $R^2 = 0.56$. The retrieval of the pond depth is more uncertain: its value can differ up to 2 times from the measured one and RMSD = 65%. This is to be expected, because the pond water depth has much less effect on the pond albedo than the underlying ice thickness. Nevertheless, the correlation for the entire dataset of the measured and retrieved pond depth values is quite high ($R^2 = 0.62$) and 70% of the retrieved values are inside the 50%-error range. The



observed scatter in the retrieval results might partly be explained by the specifics of the field measurements of the water depth and ice thickness in the melt pond: ice drillings or water depth measurements are performed at one single point of the melt pond and do not necessarily represent the average ice thickness or water depth values which can be highly variable.

Summarizing the verification, we can say that the spectra retrieval in the visible range is good for all the considered cases.

Some difference is observed in the blue, when some colored organic matter or mineral sediments are present in the melt water, and in the IR, where the reflectance is too low and the signal is noisy.

## 5   Conclusion

This work presents the optical model of melt ponds on sea ice. Similar to Makshtas and Podgorny (1996) we assume a pond

to be a plane-parallel layer of pure water on an ice substrate. We paid much attention to the pond bottom albedo as it is the main factor that determines the pond reflectance. The albedo of the ice substrate is calculated within the modified two-stream approximation (Zege et al, 1991), which relates the layer albedo to the transport scattering coefficient of the medium and its thickness. The analysis of the spectral behavior of the characteristics of the sea ice constituents (air bubbles and brine inclusions) has shown that the average cosine of the scattering phase function, and therefore the transport scattering

coefficient of sea ice, is spectrally neutral. Hence, the pond can be characterized by only three independent parameters that determine its reflectance through the visible and near IR spectral range: the pond depth, the ice substrate thickness, and the ice transport scattering coefficient. The developed model proposes the simple analytical formulas to calculate the main reflective characteristics of a melt pond: the bidirectional reflectance factor and the black and white-sky albedo. The derivation of the analytical formulas becomes possible due to the assumption of the Lambert reflection by the pond bottom.

Although this assumption does not meet the reality in general, the model verification with the field measurements approves its reasonableness, at least, concerning the spectral albedo. Its validity for the pond BRF requires further investigations.

Additional attention is paid to the correct account for the illumination conditions during the field measurements. It is shown that multiple reflections of light between the atmosphere and surrounding background can be neglected, so the *a priori* knowledge of the background albedo is not necessary. However, the sky conditions (overcast or clear, presence of cirrus or

aerosol load) should be specified to interpret the pond albedo as the white, black, or blue-sky ones. In the last case it is highly desirable to know the spectrally resolved atmospheric optical thickness for the field measurements.

The model presented was successfully used in the algorithm for the sea ice albedo and melt pond fraction retrieval from the MERIS data (Zege et al., 2015; Istomina et al., 2015a; Istomina et al., 2015b). The model provides accurate description of the melt pond reflective properties. It is robust and is able to reproduce a variety of melt pond types observed in the field.

The presented model can be useful in the problems of physics of sea ice and in monitoring the melt of the Arctic and Antarctic sea ice cover. Moreover, this makes it possible to improve the parameterization of the underlying surface in



various atmospheric remote sensing retrievals over the Arctic summer sea ice (clouds, aerosols, trace gases) and potentially re-evaluate the climatic feedbacks and radiative budget of the Arctic region at a new accuracy level.

## 6    Data availability

The field data from the R/V *Polarstern* cruise ARK XXVII/3 are available at the PANGAEA data repository (Istomina et al., 2016, 2017).
The field data from the Barrow-2008 expedition are available at the Arctic Data Center: spectral albedos – https://arcticdata.io/catalog/#view/doi:10.5065/D6NZ85TB, line photos – https://arcticdata.io/catalog/#view/doi:10.5065/D6J1019P .
The field data from the SHEBA-1998 expedition are available in a supplement to this manuscript.

## Acknowledgements

The work was supported by the Institutional Strategy of the University of Bremen, funded by the German Excellence
Initiative, and by the TR 172 "ArctiC Amplification: Climate Relevant Atmospheric and SurfaCe Processes, and Feedback Mechanisms (AC)3," funded by the German Research Foundation (DFG).

The authors are grateful to the scientific party of the ARK XVII/3 cruise for making the spectral albedo measurements possible. Special thanks are expressed to M. Nicolaus for organizing the logistics and to the Sea Ice Physics group on board
for assisting with the measurements.

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





**Table 1. Melt pond characteristics**

| Layer | Predefined characteristics | Variable characteristics |
|---|---|---|
| Air-water boundary (AW) | Spectral refractive index $n$ ; Fresnel reflectance $R^F$ and transmittance $T^F$ | No |
| Water | Water absorption coefficient $\alpha_w$ (spectral); Rayleigh scattering coefficient $\sigma_w$ (in the law of extinction only) | Pond depth $z$ |
| Substrate ice layer (pond bottom) | Ice absorption coefficient $\alpha_i$ (spectral); scattering within the two-stream and transport approximations | Transport scattering coefficient $\sigma_t$<br><br>Thickness $H$ |



**Table 2. Measure and retrieved pond geometric parameters**

| Fig | Pond codename | ice thickness (cm), retrieved | water depth (cm), retrieved | ice thickness (cm), measured | water depth (cm), measured | RMSD (total) | RMSD (visible) |
|---|---|---|---|---|---|---|---|
| **7** | 1008_P2 | 276 | 12 | 230 | 21 | $1.3 \times 10^{-2}$ | $1.0 \times 10^{-2}$ |
| | 1008_P3 | 130 | 12 | 225 | 14 | $2.2 \times 10^{-2}$ | $1.5 \times 10^{-2}$ |
| | 210812purbp1e00000 | 163 | 6 | 182 | 11 | $1.5 \times 10^{-2}$ | $1.4 \times 10^{-2}$ |
| **8** | 110812ROVtransect23e24p00000 | 182 | 29 | - | - | $1.6 \times 10^{-2}$ | $1.0 \times 10^{-2}$ |
| | 210812purbp3e00000 (2208_P3) | 212 | 24 | 143 | 30 | $1.2 \times 10^{-2}$ | $7.2 \times 10^{-3}$ |
| | 210812purwhitep4e00000 (2208_P4) | 89 | 0 | 132 | 20 | $1.5 \times 10^{-2}$ | $1.3 \times 10^{-2}$ |
| **9** | 260812Larm2pond1e00000 | 28 | 89 | - | 30 | $8.6 \times 10^{-3}$ | $5.2 \times 10^{-4}$ |
| | 260812Larm2pond2e00000 | 33 | 59 | - | 30 | $8.1 \times 10^{-3}$ | $7.1 \times 10^{-4}$ |
| | 260812purdpw3e00000 (2608_P3) | 63 | 38 | 49 | 30 | $6.6 \times 10^{-3}$ | $4.0 \times 10^{-4}$ |
| **10** | 260812purbp1e00000 (2608_P1) | 164 | 61 | 256 | 36 | $9.7 \times 10^{-3}$ | $6.0 \times 10^{-3}$ |
| | 260812purbp2de00000 | 170 | 63 | - | 50 | $1.1 \times 10^{-2}$ | $4.0 \times 10^{-3}$ |
| | 210812puralg5e00000 (2208_P5) | 15 | 22 | 33 | 20 | $6.6 \times 10^{-3}$ | $4.6 \times 10^{-3}$ |
| **11** | Barrow Blue MP #7 | 53 | 22 | - | 15 | $9.3 \times 10^{-3}$ | $1.2 \times 10^{-2}$ |
| | Barrow BubblyBlue MP #4 | 55 | 11 | - | 6 | $1.0 \times 10^{-2}$ | $1.2 \times 10^{-2}$ |
| | Barrow BlueWithBrownSpots #3 | 52 | 17 | - | 20.5 | $8.3 \times 10^{-3}$ | $1.1 \times 10^{-2}$ |
| **12** | SHEBA light Jul 17 | 72 | 38 | - | 30 | $3.2 \times 10^{-3}$ | $3.8 \times 10^{-3}$ |
| | SHEBA light Jul 21 | 70 | 38 | - | 33 | $3.7 \times 10^{-3}$ | $4.2 \times 10^{-3}$ |
| | SHEBA light Jul 25 | 41 | 45 | - | 38 | $4.9 \times 10^{-3}$ | $5.8 \times 10^{-3}$ |
| | SHEBA light Jul 29 | 44 | 51 | - | 40 | $6.5 \times 10^{-3}$ | $6.1 \times 10^{-3}$ |
| | SHEBA light Aug 2 | 48 | 49 | - | 43 | $4.7 \times 10^{-3}$ | $5.6 \times 10^{-3}$ |
| | SHEBA light Aug 6 | 6 | 72 | - | 44 | $7.4 \times 10^{-3}$ | $8.5 \times 10^{-3}$ |
| | SHEBA light Aug 10 | 9 | 68 | - | - | $7.6 \times 10^{-3}$ | $8.5 \times 10^{-3}$ |
| | SHEBA light Aug 14 | 83 | 30 | - | - | $6.4 \times 10^{-3}$ | $6.9 \times 10^{-3}$ |
| | SHEBA dark Jul 17 | 107 | 41 | - | 30 | $2.0 \times 10^{-3}$ | $2.2 \times 10^{-3}$ |
| | SHEBA dark Jul 21 | 108 | 44 | - | 33 | $2.0 \times 10^{-3}$ | $2.3 \times 10^{-3}$ |
| | SHEBA dark Jul 25 | 84 | 47 | - | 38 | $2.1 \times 10^{-3}$ | $2.4 \times 10^{-3}$ |
| | SHEBA dark Jul 29 | 68 | 67 | - | 40 | $5.3 \times 10^{-3}$ | $3.3 \times 10^{-3}$ |
| | SHEBA dark Aug 2 | 75 | 52 | - | 43 | $2.1 \times 10^{-3}$ | $2.3 \times 10^{-3}$ |
| | SHEBA dark Aug 6 | 11 | 101 | - | 44 | $4.1 \times 10^{-3}$ | $2.5 \times 10^{-3}$ |
| | SHEBA dark Aug 10 | 14 | 100 | - | - | $3.7 \times 10^{-3}$ | $1.3 \times 10^{-3}$ |
| | SHEBA dark Aug 14 | 87 | 35 | - | - | $1.8 \times 10^{-3}$ | $2.1 \times 10^{-3}$ |





**Figures**

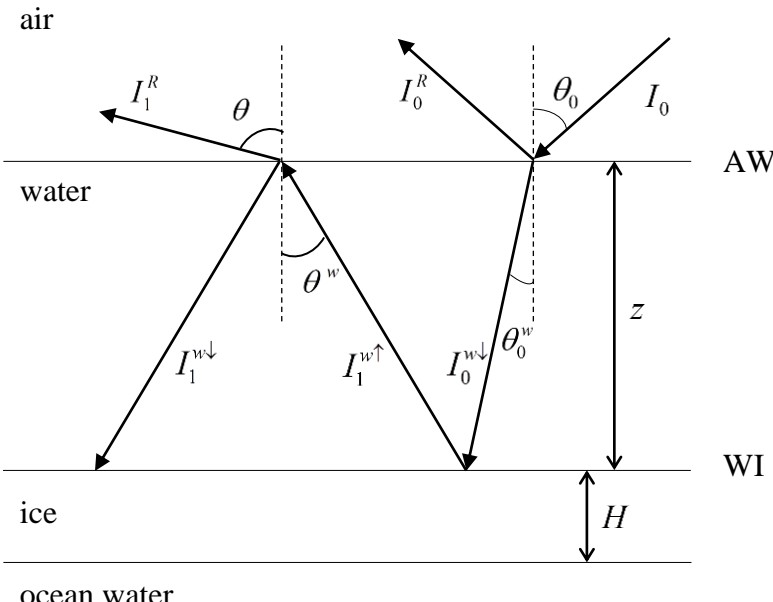

**Figure 1.** Schematic image of light rays in a melt pond. $I_0$ is the intensity of the incident light, $\theta_0$ is the angle of incidence; $I_0^R$ is the intensity of light, reflected from the air-water interface (AW); $I_0^{w\downarrow}$ is the intensity of light, refracted by the AW interface, $\theta_0^w$ is the angle of refraction; $I_1^{w\uparrow}$ is the intensity of the up-welling diffuse light, $\theta^w$ is the angle of internal reflection; $I_1^{w\downarrow}$ is the intensity of light after internal reflection by the AW interface, $\theta^w$ is the angle of internal reflection; $I_1^R$ is the intensity of light that comes out of the melt pond after refraction by the AW interface, $\theta$ is the observation angle equal to the angle of refraction.




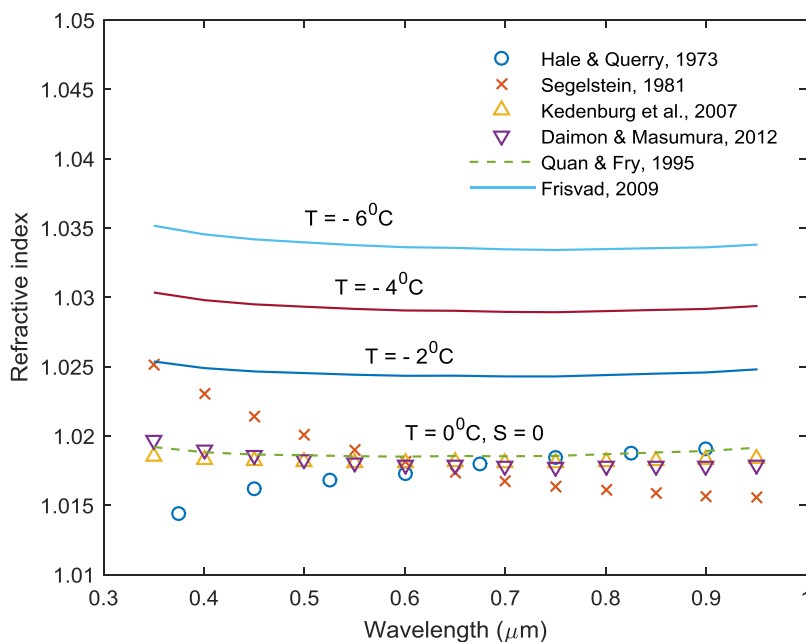

**Figure 2**. Spectra of the relative refractive index 'water-to-ice': distilled water (signs), brine with zero salinity at $0^0$C (dashes), and brine with equilibrium salinity at different temperatures (solids).



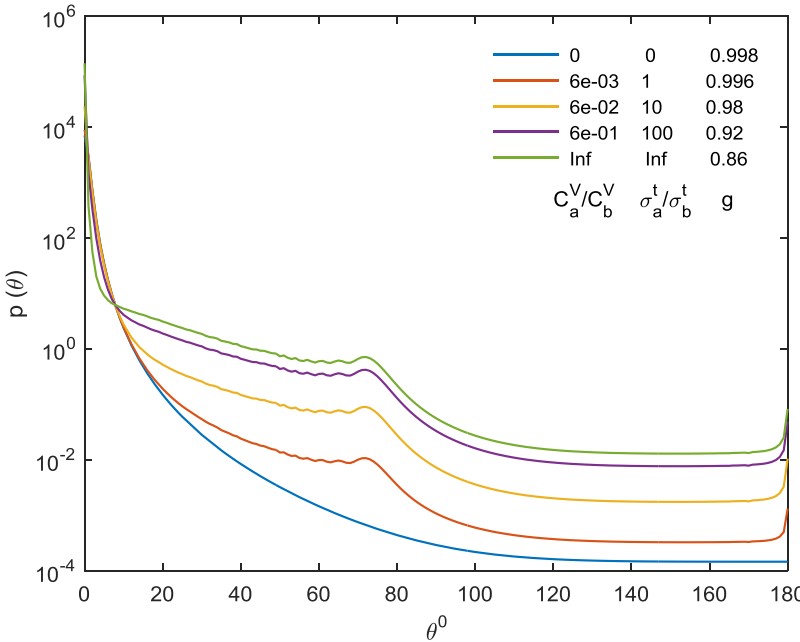

**Figure 3.** Phase functions of the mixture of air bubbles and brine inclusions at –2⁰C with different fraction ratio $C_a^V / C_b^V$. The ratio of transport scattering coefficients $\sigma_a^t / \sigma_b^t$ and the average cosine $g$ are also shown. The effective sizes are $R_a$=42.55 μm, $R_b$=100 μm.



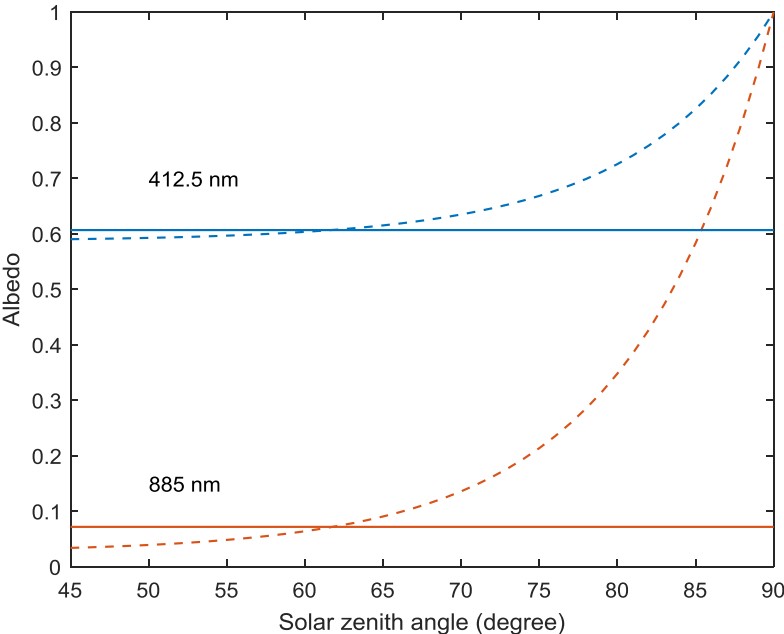

**Figure 4.** Black-sky albedo of a light melt pond ( $z = 17\text{cm}$ , $\sigma_t = 3.2\text{m}^{-1}$ , $H = 1.25\text{m}$ ) vs. the angle of incidence (dashed). The white-sky albedo values are shown in solid.




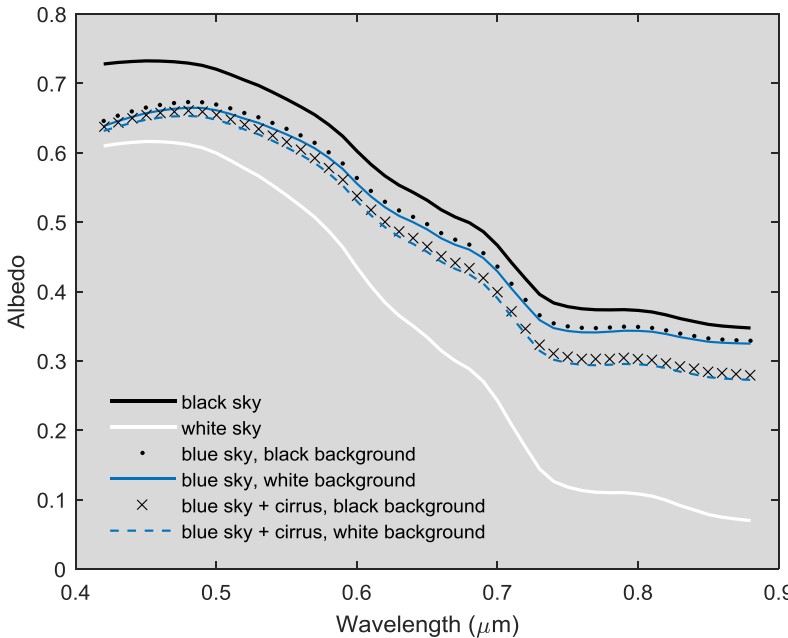

**Figure 5.** Spectra of melt pond albedos at various illumination conditions and background albedo.





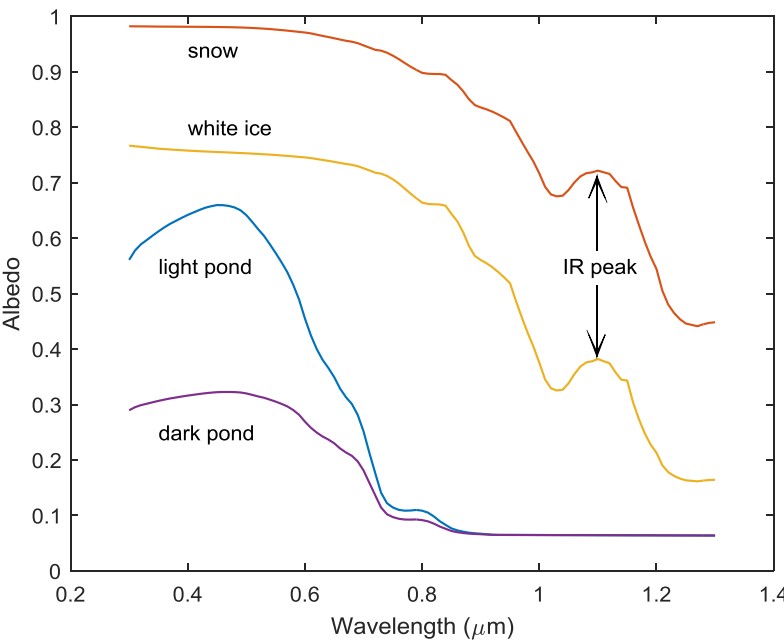

**Figure 6.** Spectral albedo of melt ponds, snow, and white ice. Light pond: depth is 20cm, $\sigma_t = 4\mathrm{m}^{-1}$, transport optical depth is 5; dark pond: depth is 20cm, $\sigma_t = 2\mathrm{m}^{-1}$, transport optical depth is 1; white ice: the effective grain size is 2mm, optical depth is 12; snow: the effective grain size is 0.2mm, optical depth is 200.





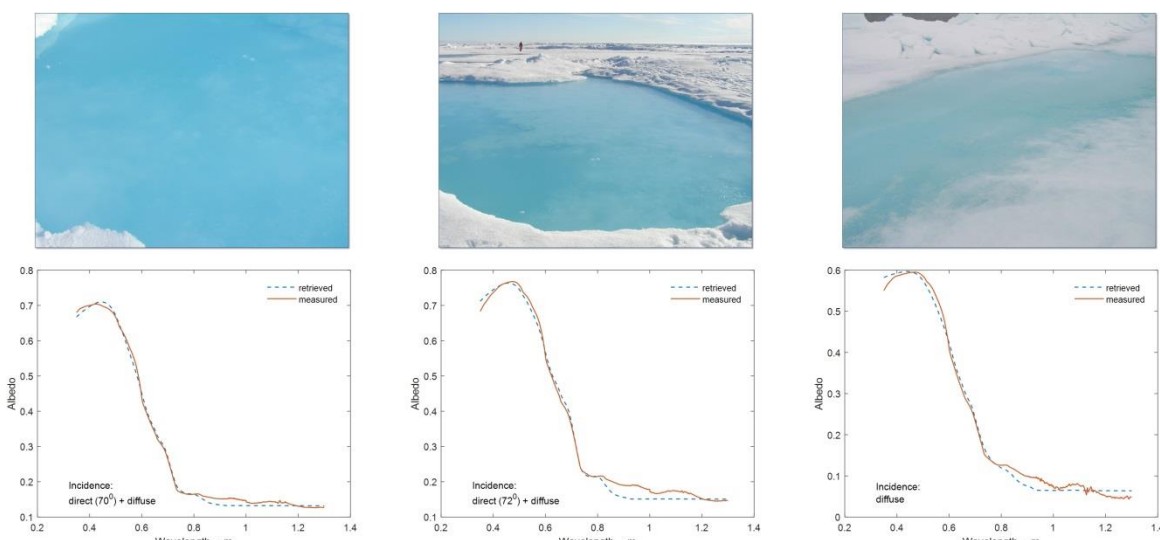

**Figure 7.** Light frozen blue ponds





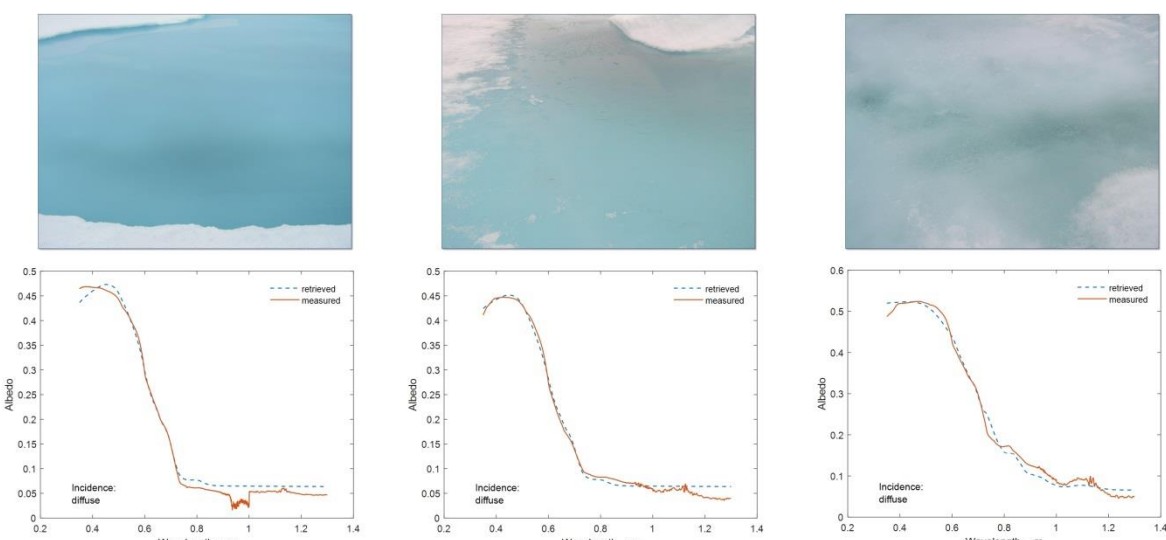

**Figure 8.** Frozen blue ponds





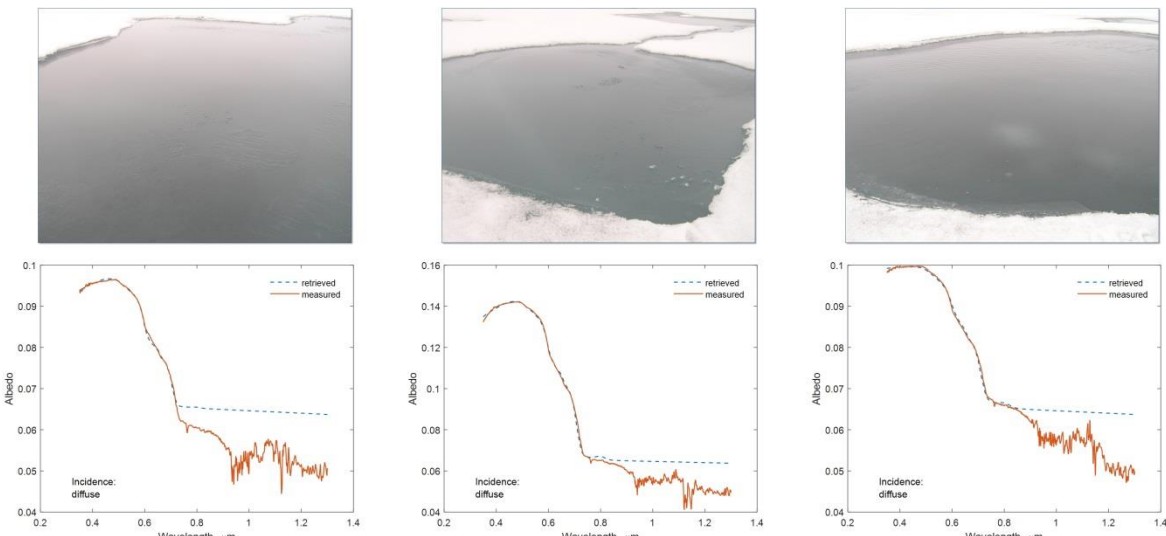

**Figure 9.** Dark open ponds





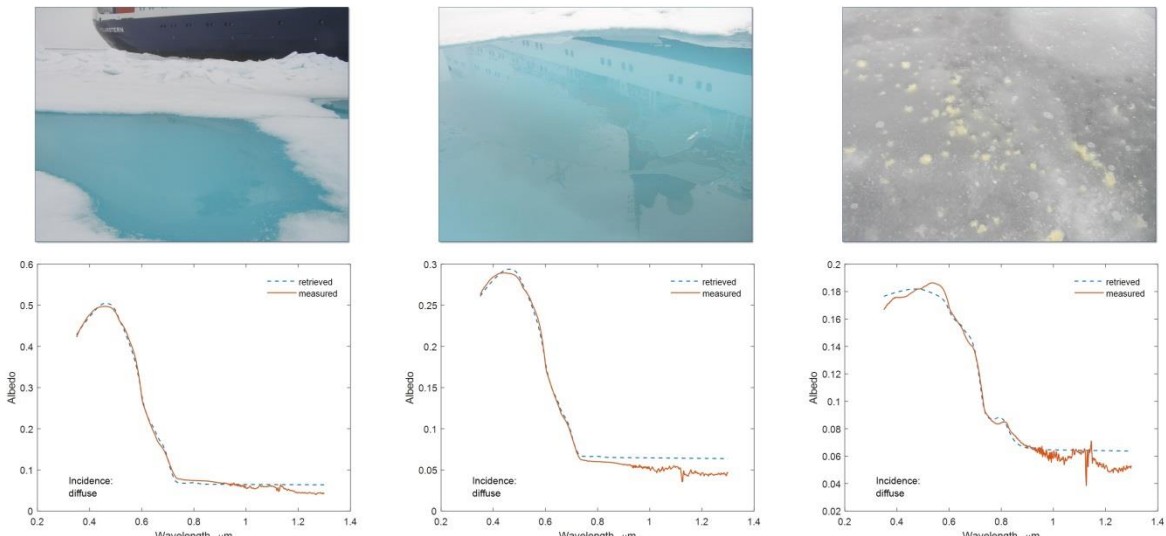

**Figure 10.** From left to right: the light blue pond, a darker part of the blue pond, and the dark pond with yellow algae




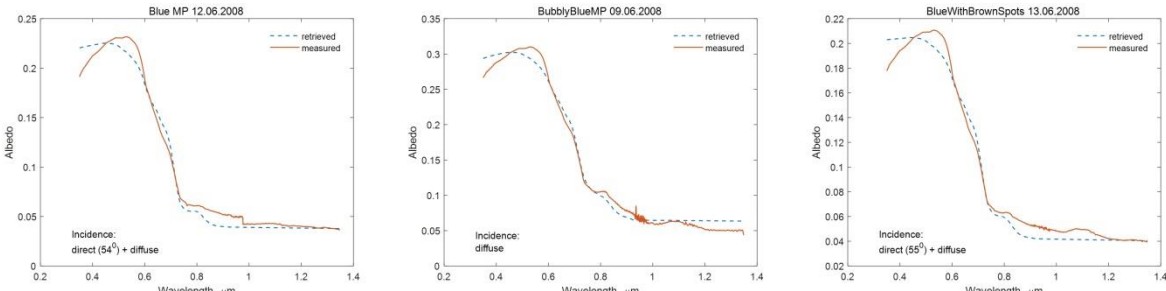

**Figure 11.** Spectra of some melt ponds from *Barrow-2008*: (left to right) a blue melt pond, a bubbly blue melt pond, and a blue melt pond with brown spots.





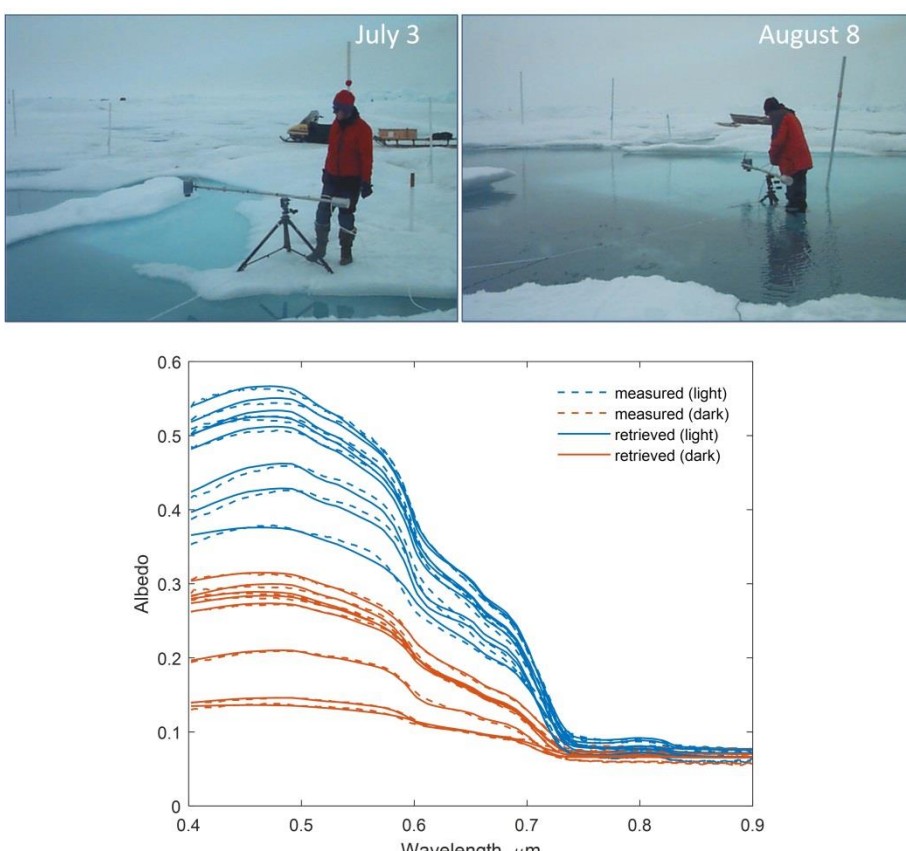

**Figure** 12. SHEBA dual pond: photos and spectra, measured (dashed) at the light (blue) and dark (red) parts and simulated (solid). The photographs are taken at the early and late melt season (on June 3 and August 8, respectively).



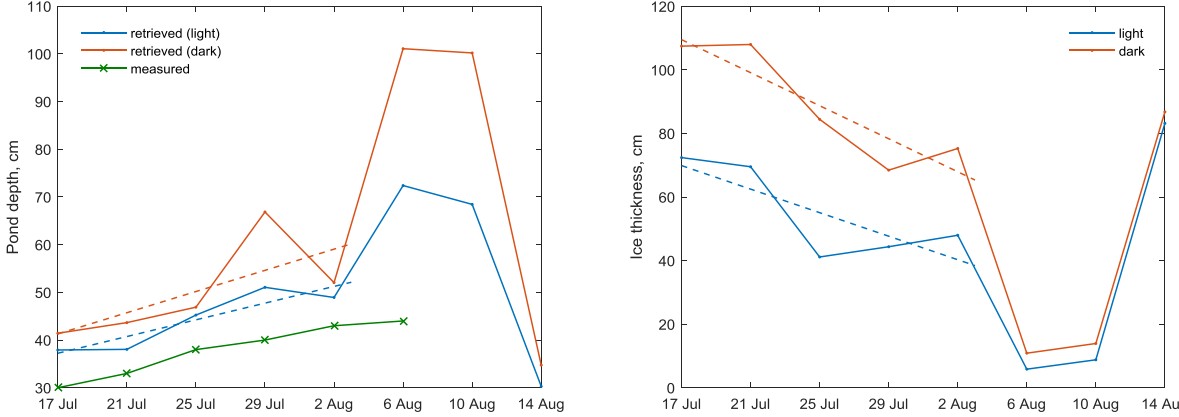

**Figure 13.** Retrieved pond depth (left) and ice thickness (right) for the two parts of the dual pond shown in Fig. 12. The measured pond depth is shown with crosses. The dashed lines show the linear regression for the first 5 dates.





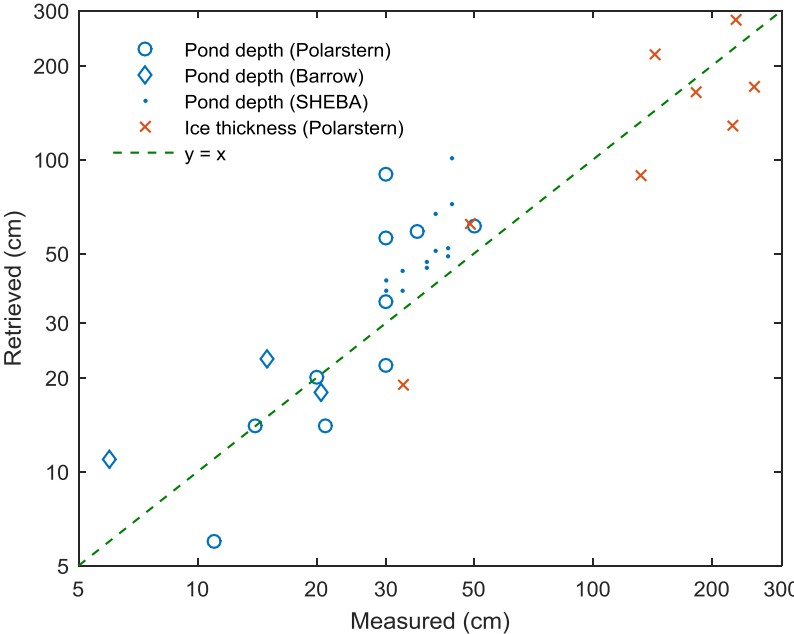

**Figure 14.** Ice thickness and pond depth, measured at different stations and retrieved. For ice thickness $R^2 = 0.56$ (N=8) and for pond depth $R^2 = 0.62$ (N=26).