# Peer review of "Reflective properties of melt ponds on sea ice"

_The Cryosphere, 2017_

## Referee Comment (RC1) · Anonymous Referee #1 · 15 Oct 2017

This manuscript details a model simulating shortwave radiative transfer for melt ponds on the surface of Arctic sea ice. The paper is of interest to TC readership and describes a model that appears sound and well tested. The language is a bit awkward in places (see minor comments below), but I do think it is generally readable.

My only major comment on the presentation is that p. 16 line 16 states that three independent parameters are required for this model: pond depth, ice substrate thickness, and ice transport scattering coefficient. I agree. The results presented in Table 2 show the first two parameters. What is assumed about the third one? There is no information discussed in the manuscript that would suggest what values were inferred for the ice transport scattering coefficient. Almost all of the comparisons between model and observation show remarkable fidelity. It seems so remarkable, that I wonder what range of transport scattering coefficients are used, and whether there is some vertical variability allowed in the ice layer beneath the pond water for that coefficient? If I un-

[Figure]

derstand correctly, the model is inferring an optical depth—so an assumption must be made about the inherent optical properties in order to retrieve the physical depth of the ice? What is that assumption?

Minor comments:

p.1 line 9, 19: "large part" and "large fraction" are not very specific

p.1 line 24: "nowadays" colloquial

p.2 line 5 -6: "Makshtas and Podgorny give a formula for pond spectral albedo at direct incidence only; they do not consider the angular distribution of the reflected light." This sentence is a bit confusing. I understand that M&P give a formula for pond albedo only for direct incidence, but I don't see why that relates at all to an angularly-resolved description of the reflected field.

p.2 line 15: "banner of the ice grains presence" makes no sense

p. 2 line 16: "common deficient information" makes no sense

p.4 line 1: define 'AW'

p.5 line 3: define 'WI'

p.6 line 6: please supply a reference for the definition of 'transport scattering coefficient'

p. 6 line10: does 'very elongated' phase function mean 'very forward peaked' phase function? I don't believe 'very elongated' is commonly understood. I think the authors are attempting to convey the idea that a smaller scattering coefficient and lower |g| can be used to describe the apparent optical properties of a medium with large scattering coefficient and/or high |g|.

p. 6 line 22: mirabilites and hydrohalites. . . should be mirabilite crystals and hydrohalite crystals

p.6 line 26-28: If the highly scattering surface layer isn't being considered here, then

what is being considered?

p.6 line 28: Statement that air bubbles in sea ice are mostly spherical needs a reference.

p.7 line 3: is exponent +1.24 or -1.24?

p.8 line 20: sloppy notation, with the 't' used as a subscript on the left hand side of the equation and as a superscript on the right hand side, but both mean the same thing.

p. 11 line 11: 'extra-terrestrial solar irradiance' I think is more commonly called 'top-of-atmosphere irradiance'? eqn 49: it is confusing that both A and alpha are used for albedo

p. 13 line 9 – 10: melt ponds forming during 2 Aug – 8 Oct cruise? Seems unlikely.

p. 13 line 11-14: The description here lacks detail. I assume the fiber optic probe coupled to the ASD is used to view light reflected by the Spectralon plate, but this isn't adequately described. The phrase "served as a diffuser" doesn't completely describe how the Spectralon plate was employed.

p. 13 line 31: what does 'open' mean here? No ice skim?

p. 14 line 31: the spectral albedo was taken every 4 days?

Fig 5 The angle of incidence is stated in the text, but needs to also be stated in the figure caption.

Fig 6 Where did these spectral curves come from? There needs to be some data attribution.

Fig 7 caption should include information (from text) that these all had 2-3 cm layer of ice on top.

Fig 7 I am surprised at how high the albedo is at blue wavelengths! Could this be due to the frozen surface? If so, then that would contradict the statement p.13 line 21. I would

expect the peak albedo at blue wavelengths for unfrozen melt ponds to be somewhere in the range 0.1 to 0.5, at most. Would be useful to show all the panels in each cluster (Figs 7, 8, 9 each a cluster) on the same vertical (albedo) scale. Also, captions for Figs 7, 8, 9, 10, 11 need to contain information about the general locations of each series.

Fig 8 If these ponds were heterogeneous, then the exact location of the albedo measurement matters! Can this location be shown?

Fig 12 Caption says 'on June 3', but I believe was July?

---

## Referee Comment (RC2) · Anonymous Referee #2 · 17 Jan 2018

The manuscript describes a new numerical model to calculate the spectral reflectance of melt ponds on Arctic sea ice, mostly determined by three independent variables. The authors find good agreement between simulated and observed spectra from in-situ measurements during three different field campaigns. This allows them to derive water depth, under pond ice thickness and transport coefficients for each of the ponds. Given the ongoing changes of the Arctic sea ice cover towards longer melt periods and increasing melt pond fractions, the manuscript describes a timely topic, which is well suited for publications in The Cryosphere.

Over all, I suggest publication after minor revisions, which mostly comprise some additional discussion and sharpening of the main conclusions.

General comments:

- It is not clear to me what the NEW elements of this model are, compared to existing

models and theoretical approaches. It seems that most relations and assumptions are taken from existing studies. Since this is a mostly methodological manuscript, the following aspects need to become obvious:

o What are the additional and new insights into radiative transfer of melt ponds?

o How can or should this model be used in future (the outlook at the very end is rather unspecific and too general)?

o What kind of scientific merit do the authors expect from this and following studies (applications of the model).

- The authors conclude that only three independent parameters are needed to characterize melt ponds and thus to retrieve an appropriate optical characterization from them. They do discuss and show results of pond depth and substrate thickness, but I am missing an analysis and more discussion and details on the transport coefficient. In that respect, the role of the three main parameters should be discussed in the discussion and be concluded at the end of the manuscript. How do they impact the model (not only in equations) and what sensitivity do we expect and receive?

- The comparison with in-situ observations show differences of under-pond ice thickness and water depth of 50% and some even significantly higher. I do not follow the argumentation that this is satisfactory, in particular since there is very little discussion about this (see comments below). I consider these differences as more significant than the discussion reveals. In particular with respect to the under-pond (substrate) thickness, which should be the most important parameter to determine pond albedo. Note: I am puzzled about the term "substrate". Why not under-pond ice thickness?

Specific comments:

Abstract:

The abstract may be significantly improved by adding more results and a statement that explicitly names the additional benefit and further applications of the model:

- Page1/Line15 (P1/L15): . . . are examined: What is the result of the examination?

- P1/L16: several => three field campaigns

- P1/L17: "good performance" this is rather relative, good in what measure?

- Why are the three main parameters not mentioned in the abstract? How do they perform?

- What does this model stand out for and what is the (likely) future benefit of this study/model?

Introduction

- Recent studies by different groups show the increasing fraction and importance of melt ponds. Also shifts in melt onset and melt season duration are observed and discussed in various ways. I am missing this aspect in the introduction, while this would add to the motivation of this study and model development.

- In addition, there are various approaches to parameterize melt ponds in circulation models of various complexities. This should also be included and could even link to the role of light transmittance into and through sea ice (the remaining after reflection). This could also well link the introduction to the final part of the conclusions (see comments below)

- P2/L4: Include also "water" properties.

Model descriptions

- This section is most detailed. It could be improved by distinguishing better between existing models and theories and highlighting new ideas and findings.

- The role of the resulting three main parameters should be highlighted.

- It would add value to the manuscript if the model is made available for other users. How is the model implemented? How (numerically) costly are the simulations?

Model verification

- P13/L16-19: The realization of the validation and comparison should be described in more detail.

o How did the authors derive that these are the three main parameters. What other parameters were analyzed?

o What about the transport coefficients? How were they studied/discussed?

o How are the thicknesses retrieved?

- It is a disadvantage that most ponds were not open ponds as it is assumed in the model development. I do see the constrains through the given data set, but this weakens the verification and needs more consideration. Why is there e.g. no thin surface ice in the model?

- P14/L14: Add the year (2008) into the main text.

- Section 4.4 should be the main discussion of the comparisons. This is too short and somewhat superficial.

o Where do these rather large differences of 50% come from? I do see various reasons in e.g. pond depth distributions, non-planar interfaces, footprint of sensors compared to pond properties. But this needs to be discussed in more detail.

o What precision may/can be expected in such models?

o What determines the uncertainties? Which of the given assumptions might not be ideal, but what would it mean to adapt this? It is most likely not realistic within this study, but some additional discussion would be useful and interesting for further studies.

- With respect to those differences: As discussed, impurities are mostly low in the ponds, so the result is mostly based on scattering (not absorption). In this case, the retrieved spectral shape may be expected to be in good agreement, while amplitude is

the main aspect of evaluation. But if then the simulated differences are still around 50% for the under-ice thickness this is somewhat surprising to me. I agree that the RMSE match is quite good if not excellent, but may be not because of the right thicknesses, but other parameters in the model. This should be discussed more.

Conclusions

- Given that ponds may be described by the three parameters: How would future applications look like? What is the main benefit from this conclusion? (P16/L16)

- P16/L27: This raises the question: How much of the model has been used before and what is new (see above)?

- P16/L30: "can be useful": This is somewhat vague. How can it realistically be used?

- The last lines of the manuscript are not convincing to me. How would these improvements be implemented? What are the next applications or which part of these results is most promising. This needs a more thoroughly discussion and a more specific outlook.

- The conclusions section misses a conclusion on the uncertainties and deviations from the field measurements (Section 4.4). At the same time, I suggest to highlight that the validation was done against quite a suite of field measurements and variable pond conditions. This is a valuable aspect and could be stressed more. Many studies limit their validation to a single data set (e.g. one field experiment).

Table 1

- I think that this is not needed.

Table 2

- The pond code names seem to be an internal coding with almost no use for other studies. Using station names and dates as identifiers that link to field reports, Polarstern station numbers, and Pangaea data sets is suggested.

- I suggest to re-arrange the columns and group retrieved/measured/difference (absolute, and %) for each: ice thickness and water depth. This eases evaluation of the performance.

- RMSD values could be given in units of e.g. 10ˆ-3 to save space and ease reading

Please also note the supplement to this comment:
https://www.the-cryosphere-discuss.net/tc-2017-150/tc-2017-150-RC2-supplement.pdf

---

## Editor Comment (EC1) · J.-L. Tison (Editor) · 17 Jan 2018

Dear Authors,

We have now received the comments of the two referees and both recommend publication with minor changes, which is good news. I would first like to apologize for the delays encountered during this phase. Expert are difficult to find, and those who were willing to review the manuscript had long engagements in the field. Also, the Christmas/New Year period is not very favourable to swift action!..

Although both reviewers qualify the revision as "minor", I still believe that there is a fair amount of work to address their comments. For example, both reviewers noted the absence of discussion about the third controlling parameter (transport coefficient) and reviewer #2 has the same concern as my original one, the Novelty of the paper...this needs to be extremely clearly underlined in the new version of the manuscript, since it

is a priority criteria for publication in TC!..

I would therefore strongly recommend that you address all (both general and detailed) comments raised by each reviewer in a "reply to comments" document. The easiest way for me to judge of the adequacy of your answers is to provide me with (upload) the following documents:

a) Your "reply to comments" document in which you paste the comments of REv1 and 2, and clearly give your answer underneath each of these comments (different paragraph, different color, different police..your choice). That answer should refer to the line numbers in the new version of the manuscript (see b) below) were you address the comment, when applicable

b) a new version of the manuscript (text + figures) clearly showing (e.g. highlighting) the changes made in response to the reviewers comments

c) a "clean" (no comments) new version of the manuscript (text + figures)

Good luck with it,

Best regards,

Jean-Louis Tison

---

## Author Comment (AC1) · 14 Feb 2018

We are grateful to the referees for their positive evaluation of our work and particularly for the detailed comments. We made corrections in the manuscript according to the referees' minor comments. In the following we give more detailed answers to their questions. The revised version of the manuscript will be submitted when we will have the answers of the reviewers.

Anonymous Referee #1:

This manuscript details a model simulating shortwave radiative transfer for melt ponds on the surface of Arctic sea ice. The paper is of interest to TC readership and describes a model that appears sound and well tested. The language is a bit awkward in places (see minor comments below), but I do think it is generally readable.

[Figure]

Thank you. As for our English, we did our best and particularly mindfully considered your minor comments and made appropriate corrections in the text. For the final version we will have an additional round of correction by our native speaking coauthors. Besides, there will be English copy-editing by the editorial staff at the final stage.

My only major comment on the presentation is that p. 16 line 16 states that three independent parameters are required for this model: pond depth, ice substrate thickness, and ice transport scattering coefficient. I agree. The results presented in Table 2 show the first two parameters. What is assumed about the third one? There is no information discussed in the manuscript that would suggest what values were inferred for the ice transport scattering coefficient. Almost all of the comparisons between model and observation show remarkable fidelity. It seems so remarkable, that I wonder what range of transport scattering coefficients are used, and whether there is some vertical variability allowed in the ice layer beneath the pond water for that coefficient? If I understand correctly, the model is inferring an optical depth $\tau$ so an assumption must be made about the inherent optical properties in order to retrieve the physical depth of the ice? What is that assumption?

You are absolutely right that just the optical depth, rather than the geometrical one, determines the reflectance. They are related by Eqs. (42). We consider all three parameters, z, H, and $\sigma t$, as independent ones. We vary all of them independently when fitting spectra and don't make any additional assumptions about $\sigma t$. (Except vertical variability). Of course, we don't have enough information to retrieve the vertical profile of $\sigma t$, so we assume that we retrieve some constant effective value for a layer). Thus, all these three values are retrieved for every spectrum. In Table 2 we show only two of them just for comparison with the in situ measured values of z and H. This information will also be added to the manuscript. However, nobody measures $\sigma t$, so we don't show its values. But we added the retrieved values of $\sigma t$ for the light and dark portions of the SHEBA pond (see the last paragraph of Sec. 4.3), where they are important for calculation of the scattering coefficient by bubbles.

Minor comments: p.1 line 9, 19: "large part" and "large fraction" are not very specific

We omit general references and put: "up to 60% on multiyear ice according to Maykut et al. (1992) and up to 80% on landfast ice according to Langleben (1971)."

p.1 line 24: "nowadays" colloquial

Changed: "in light of the environmental changes observed recently"

p.2 line 5 -6: "Makshtas and Podgorny give a formula for pond spectral albedo at direct incidence only; they do not consider the angular distribution of the reflected light." This sentence is a bit confusing. I understand that M&P give a formula for pond albedo only for direct incidence, but I don't see why that relates at all to an angularly-resolved description of the reflected field.

We changed the phrase to: "Besides, the question of the angular distribution of light reflected by a melt pond is still open."

p.2 line 15: "banner of the ice grains presence" makes no sense

Changed to "evidence of the ice grains presence"

p. 2 line 16: "common deficient information" makes no sense

Changed to "when the incident angle is unknown"

p.4 line 1: define 'AW' p.5 line 3: define 'WI'

Definitions are added in p.3, l.27.

p.6 line 6: please supply a reference for the definition of 'transport scattering coefficient'

We added the references Davison, 1958 and Chandrasekhar, 1960

p. 6 line10: does 'very elongated' phase function mean 'very forward peaked' phase function? I don't believe 'very elongated' is commonly understood. I think the authors are attempting to convey the idea that a smaller scattering coefficient and lower |g| can

be used to describe the apparent optical properties of a medium with large scattering coefficient and/or high |g|.

Yes, you are absolutely right. Corrected.

p. 6 line 22: mirabilites and hydrohalites. . . should be mirabilite crystals and hydrohalite crystals

Corrected.

p.6 line 26-28: If the highly scattering surface layer isn't being considered here, then what is being considered?

We tried to specify a little: "We do not consider here the highly scattering surface layer that is formed on top of sea ice during the water drainage process and is commonly referred to as 'white ice'." Hope it's clearer.

p.6 line 28: Statement that air bubbles in sea ice are mostly spherical needs a reference.

We added Gavrilo and Gaitskhoki, 1970; Mobley et al., 1998; Light, 2010.

p.7 line 3: is exponent +1.24 or -1.24?

Corrected to −1.24.

p.8 line 20: sloppy notation, with the 't' used as a subscript on the left hand side of the equation and as a superscript on the right hand side, but both mean the same thing.

We hope this notation will not confuse our readers. After all, these sub- and superscripts are not the tensor indices where their position is principal.

p. 11 line 11: 'extra-terrestrial solar irradiance' I think is more commonly called 'top-of-atmosphere irradiance'?

Both terms are widely used. As for our experience, the term 'top-of-the-atmosphere' is more frequent for the Earth reflected radiance, while 'extra-terrestrial' for solar light.

eqn 49: it is confusing that both A and alpha are used for albedo

We replaced $\alpha$ by Ablue.

p. 13 line 9 – 10: melt ponds forming during 2 Aug – 8 Oct cruise? Seems unlikely.

These are the dates of the cruise. We added: "The melt ponds were observed in August."

p. 13 line 11-14: The description here lacks detail. I assume the fiber optic probe coupled to the ASD is used to view light reflected by the Spectralon plate, but this isn't adequately described. The phrase "served as a diffuser" doesn't completely describe how the Spectralon plate was employed.

We added: "A sensor measures the light signal supplied by a fiber optical probe, which collects light reflected by a 10x10 cm2 Spectralon white plate."

p. 13 line 31: what does 'open' mean here? No ice skim?

Yes. We put an explanation in the beginning of Sec. 4.1. "The melt ponds were observed in August, being both open (with no ice skim) and frozen over (with a skim of ice), sometimes snow covered."

p. 14 line 31: the spectral albedo was taken every 4 days?

Yes. We put: "The spectra were taken every four days during this period. The spectra processing results are shown in Figs. 12 and 13."

Fig 5 The angle of incidence is stated in the text, but needs to also be stated in the figure caption.

Done.

Fig 6 Where did these spectral curves come from? There needs to be some data attribution.

These spectral curves are modeled for the typical values. We put a phrase "Typical

spectral albedo of melt ponds, snow, and white ice, calculated for the following parameters:

Fig 7 caption should include information (from text) that these all had 2-3 cm layer of ice on top.

Done.

Fig 7 I am surprised at how high the albedo is at blue wavelengths! Could this be due to the frozen surface? If so, then that would contradict the statement p.13 line 21. I would expect the peak albedo at blue wavelengths for unfrozen melt ponds to be somewhere in the range 0.1 to 0.5, at most.

Actually, we cannot be sure that high albedo values do not come from the frozen surface. If an ice skim contains a lot of air bubbles, it can increase the reflectance, but in this case it becomes indistinguishable from the ice substrate. So the optical thickness retrieved is the total thickness (skim + substrate). As we wrote, our model does not consider such cases. The statement p.13 line 21 only states that a layer of transparent ice does not change pond reflection. On the other hand, there is no restriction of 0.5 for open pond albedo. To be objective, we put the phrase in p.13, line 30: "The albedo values are extraordinarily high. This could be related with the fact that the ponds are frozen over with a 2-3 cm layer of ice on top."

Would be useful to show all the panels in each cluster (Figs 7, 8, 9 each a cluster) on the same vertical (albedo) scale.

The plots are quite small, we think it's better when the drawings take all the scale.

Also, captions for Figs 7, 8, 9, 10, 11 need to contain information about the general locations of each series.

We put in the text about the Polarstern cruise: "The stations, where ponds were observed, were located from 84°3N, 31°7E to 82°54N, 129°47E. For more information about the cruise, see Boetius et al. (2012) and Istomina et al. (2016)." For Barrow and

[Figure]

SHEBA the locations are given: Chukchi and Beaufort seas.

Fig 8 If these ponds were heterogeneous, then the exact location of the albedo measurement matters! Can this location be shown?

The exact point of the measurement can be seen in the photograph, where a person is taking observation from the light portion of the pond. Unfortunately, there is no photo for the dark one.

Fig 12 Caption says 'on June 3', but I believe was July?

Of course, July. Thank you.

---

## Author Comment (AC2) · 14 Feb 2018

We are grateful to the referees for their positive evaluation of our work and particularly for the detailed comments. We made corrections in the manuscript according to the referees' minor comments. In the following we give more detailed answers to their questions. The revised version of the manuscript will be submitted when we will have the answers of the reviewers.

Anonymous Referee #2:

The manuscript describes a new numerical model to calculate the spectral reflectance of melt ponds on Arctic sea ice, mostly determined by three independent variables. The authors find good agreement between simulated and observed spectra from in-situ measurements during three different field campaigns. This allows them to derive water depth, under pond ice thickness and transport coefficients for each of the ponds.

[Figure]

Given the ongoing changes of the Arctic sea ice cover towards longer melt periods and increasing melt pond fractions, the manuscript describes a timely topic, which is well suited for publications in The Cryosphere. Over all, I suggest publication after minor revisions, which mostly comprise some additional discussion and sharpening of the main conclusions.

Thank you.

General comments: - It is not clear to me what the NEW elements of this model are, compared to existing models and theoretical approaches. It seems that most relations and assumptions are taken from existing studies. Since this is a mostly methodological manuscript, the following aspects need to become obvious: o What are the additional and new insights into radiative transfer of melt ponds?

Actually, we don't know any existing models or theoretical approaches that relate the pond reflectance to its physical characteristics. The work of Makshtas and Podgorny relates the pond albedo to the albedo of its bottom only. In our work we show how to obtain the spectrum of the pond bottom albedo through the radiative transfer characteristics of under-pond sea ice. To do so we use the approaches, developed by the authors for light scattering by non-spherical particles within the WKB approximation (Malinka, 2015) and for radiative transfer within the two-stream approximation (Zege et al, 1991). We show which particular parameters determine the pond bottom albedo. These parameters are really the transport scattering coefficient and ice thickness. Besides, we pay particular attention to two more points: the bi-directional reflectance, which is of great importance for remote sensing techniques when processing satellite data, and the atmospheric correction of in situ measured data, which is hardly made by anyone for in situ measurements. As far as we're concerned, we think that all these points are stated in the Introduction. Also, according to your advice, we added these points to the Abstract.

o How can or should this model be used in future (the outlook at the very end is rather

unspecific and too general)? o What kind of scientific merit do the authors expect from this and following studies (applications of the model).

Of course, we cannot predict all possible merits. But some applications are obvious: such a model is absolutely necessary for satellite data processing in remote sensing of Arctic ice. Particularly, this model has served as a basis for the MPD (Melt Pond Detector) algorithm for melt pond fraction and sea ice albedo retrieval from MERIS data (Zege et al., 2015).

- The authors conclude that only three independent parameters are needed to characterize melt ponds and thus to retrieve an appropriate optical characterization from them. They do discuss and show results of pond depth and substrate thickness, but I am missing an analysis and more discussion and details on the transport coefficient. In that respect, the role of the three main parameters should be discussed in the discussion and be concluded at the end of the manuscript. How do they impact the model (not only in equations) and what sensitivity do we expect and receive?

We consider all three parameters, z, H, and $\sigma$t , as independent ones. All these three values are retrieved for every spectrum. In Table 2 we show only two of them just for comparison with the in situ measured values of z and H. Nobody measures $\sigma$t , so we don't show its values. Additionally, we can add that the transport scattering coefficient is mostly variable due to air bubbles in sea ice. We appended the section dedicated to the dual pond measured in SHEBA expedition with the transport coefficient values for the light and dark parts, which gives the idea of the effect of the transport coefficient on the pond albedo.

- The comparison with in-situ observations show differences of under-pond ice thickness and water depth of 50% and some even significantly higher. I do not follow the argumentation that this is satisfactory, in particular since there is very little discussion about this (see comments below). I consider these differences as more significant than the discussion reveals. In particular with respect to the under-pond (substrate)

thickness, which should be the most important parameter to determine pond albedo.

Actually, the most important parameter that determines the pond albedo is the transport optical thickness of under-pond ice $\tau_t$ that is a product of the transport scattering coefficient $\sigma_t$ and ice thickness H: $\tau_t = \sigma_t H$. Partially this explains the retrieval error: $\tau_t$ is retrieved with much higher accuracy, however there is no way to compare it with a measured value. There could be also other different sources of error. First, the under-pond ice might not be flat, especially its lower boundary. In this case the optical retrieval gives some average value, while the in situ measurement gives a random value taken in some particular point. From this point of view the measurement makes a mistake, rather than the retrieval. The second source can be the presence of some impurities that affect the absorption spectrum. Additional absorption can affect the retrieval of the scattering coefficient and, consequently, of H. Besides, there could be other sources of uncertainties, like finite pond size, presence of snow in the receiver FOV, clouds in the sky etc. In view of that, the RMS error of 37% seems to us more than reasonable, especially given the fact that the microwave sounding methods fail absolutely in ice thickness retrieval, when ice is covered with a thin water layer.

Note: I am puzzled about the term "substrate". Why not under-pond ice thickness?

Thank you for the prompt. As we already mentioned, we are not native English speakers. We have changed this term.

Specific comments: Abstract: The abstract may be significantly improved by adding more results and a statement that explicitly names the additional benefit and further applications of the model: - Page1/Line15 (P1/L15): . . . are examined: What is the result of the examination?

We added: "We find that atmospheric correction is necessary even for in situ measurements. Thus, an atmospheric correction procedure has been used in the model verification"

- P1/L16: several => three field campaigns

Changed

- P1/L17: "good performance" this is rather relative, good in what measure?

How can we measure the adequacy of a model or a theory? This is rather quality, than quantity measure.

- Why are the three main parameters not mentioned in the abstract? How do they perform?

We added some details into the Abstract. Now they are mentioned.

- What does this model stand out for and what is the (likely) future benefit of this study/model?

The model is needed to get and study a quantitative relationship between the physical characteristics of a melt pond and its reflectance. This quantitative characterization will be helpful in retrieving melt pond fraction from space and thereby quantifying the atmosphere–sea ice–ocean heat fluxes relevant for climate research.

Introduction - Recent studies by different groups show the increasing fraction and importance of melt ponds. Also shifts in melt onset and melt season duration are observed and discussed in various ways. I am missing this aspect in the introduction, while this would add to the motivation of this study and model development.

We added these facts into the Introduction, together with the reference 'Markus et al., 2009'.

- In addition, there are various approaches to parameterize melt ponds in circulation models of various complexities. This should also be included and could even link to the role of light transmittance into and through sea ice (the remaining after reflection). This could also well link the introduction to the final part of the conclusions (see comments below)

We added the phrase about light transmittance to the Conclusion

- P2/L4: Include also "water" properties.

If we understand correctly, this comment refers to the sentence "This solution has required the detailed consideration of the inherent optical properties of sea ice, which forms the pond bottom." If so, we don't think it is worth including 'water properties', because this would mean 'sea water' IOPs, which is a very elaborated problem that is very separate from 'sea ice' IOPs.

Model descriptions - This section is most detailed. It could be improved by distinguishing better between existing models and theories and highlighting new ideas and findings.

It is stated in the Introduction: "Subsection 2.1 presents the derivation of the formulas for pond reflectance, given by Makshtas and Podgorny (1996) expanded to various incident conditions." All other findings throughout the manuscript are original. We do not see how to distinguish better.

- The role of the resulting three main parameters should be highlighted.

These parameters determine the pond spectral reflectance. The coincidence of measured and modeled spectra allows us to state that on more parameters can improve the model and make it closer to reality (unless we see real difference in spectra, which we attribute to some sediments). (we added this to conclusion, also, see below). Additionally, we added the explanation to the end of Sec. 2.4: "So, in the absence of pollutants just three parameters determine the pond spectral reflectance: namely, the transport scattering coefficient and geometric thickness of the under-pond ice and water layer depth . This statement is confirmed by the coincidence of measured and modeled spectra demonstrated below."

- It would add value to the manuscript if the model is made available for other users. How is the model implemented? How (numerically) costly are the simulations?
The model is very simple in the implementation, because it is entirely based on analytical formulas. The only numerical cost is the calculation of functions fin and fout (integrals in Eq. (14) and (22)). However, these functions can be calculated once for given set of wavelengths and then used as a look-up-table to speed up the simulation. As for the rest, all the formulas are given in the manuscript and can be used straightforwardly. We added this aspect to the end of Sec.2.1 and to the Conclusion.

Model verification - P13/L16-19: The realization of the validation and comparison should be described in more detail.

To find the best fit solution we use the multidimensional Newton-Raphson method with the singular value decomposition of the pseudo-inverse matrix. We really think that the discussion of the method lies far beyond the paper scope, but the method name is added to the manuscript. Adding computational details will make the understanding of the work only harder. Also we are sure that the particular method of searching solution doesn't matter for model verification. It is enough that we find such values of the three pond parameters that give the best fit of spectra in the sense the least squares.

o How did the authors derive that these are the three main parameters. What other parameters were analyzed?

See above our answer about the role of these three parameters. Additionally we can note that refractive indices and absorption spectra of ice and water were not analyzed, because they are fixed, and sediment concentration was not analyzed, because we have no information about polluting substances. So, no more parameters can be analyzed from the point of view of albedo spectrum. Another question is that the transport scattering coefficient consists of the contributions of air bubbles and brine inclusions and thus is determined by their concentrations. Their relationships are considered in detail in Sec. 2.2c and 4.3.

o What about the transport coefficients? How were they studied/discussed? o How are the thicknesses retrieved?

[Figure]

All three parameters are retrieved in the same manner. They comprise a 3d-vector, which is varied to provide the best fit of spectra. We added this phrase to Sec. 4.1.

- It is a disadvantage that most ponds were not open ponds as it is assumed in the model development. I do see the constrains through the given data set, but this weakens the verification and needs more consideration. Why is there e.g. no thin surface ice in the model?

For the same reason we are also not quite satisfied with the dataset, but that's what we have. We made computations for the model with frozen surface. Adding a thin ice layer on top changes almost nothing in the results however makes formulas much more tremendous, so we decided not to include them into the manuscript. This overloaded model was formulated in our internal report. At first, we planned to attach this report to the manuscript as a supplement, however the editorial refused it. And we agree with them, because it gives too little new information.

- P14/L14: Add the year (2008) into the main text.

Added.

- Section 4.4 should be the main discussion of the comparisons. This is too short and somewhat superficial. o Where do these rather large differences of 50% come from? I do see various reasons in e.g. pond depth distributions, non-planar interfaces, footprint of sensors compared to pond properties. But this needs to be discussed in more detail. o What precision may/can be expected in such models? o What determines the uncertainties? Which of the given assumptions might not be ideal, but what would it mean to adapt this? It is most likely not realistic within this study, but some additional discussion would be useful and interesting for further studies.

Throughout the manuscript, making the derivations, we stated the assumptions we use in the model. Surely, every assumption is some approximation or idealization and any of them can limit applicability and accuracy of results. However, the perfect fit of the

measured and modeled spectra is a proof that these assumptions were reasonable.

- With respect to those differences: As discussed, impurities are mostly low in the ponds, so the result is mostly based on scattering (not absorption). In this case, the retrieved spectral shape may be expected to be in good agreement, while amplitude is the main aspect of evaluation. But if then the simulated differences are still around 50% for the under-ice thickness this is somewhat surprising to me. I agree that the RMSE match is quite good if not excellent, but may be not because of the right thicknesses, but other parameters in the model. This should be discussed more.

We think this question is answered in the section 'General comments' (the 4th question). (Also note that the mean error for ice thickness is 37%, not 50%).

Conclusions - Given that ponds may be described by the three parameters: How would future applications look like? What is the main benefit from this conclusion? (P16/L16)

It is just a scientific statement. Actually, reducing the number of key physical parameters down to three is indeed the main benefit.

- P16/L27: This raises the question: How much of the model has been used before and what is new (see above)?

This model was almost fully used in the MPD algorithm described in Zege et al., 2015, but a detailed description hasn't been published until now. The new modification is that the two-stream approximation is used now instead of the radiative transfer asymptotic formulas for weak absorption. This allows widening the scope of the applicability to significant absorption, what is important in the red and near IR range. The second one is that the scientific justification is given for the sea-ice IOPs and, consequently, to the role of the transport scattering coefficient.

- P16/L30: "can be useful": This is somewhat vague. How can it realistically be used?

For example, for a better understanding of the Arctic energy budget the quantitative characterization of melt pond reflection is needed. At least, it is needed for satellite

retrieval of melt pond fraction.

- The last lines of the manuscript are not convincing to me. How would these improvements be implemented? What are the next applications or which part of these results is most promising. This needs a more thoroughly discussion and a more specific outlook.

The most promising is the relationship between the physical and optical parameters of a melt pond. We think this relationship is needed to study, e.g., the process of ice melting, which is highly determined by its radiative budget.

- The conclusions section misses a conclusion on the uncertainties and deviations from the field measurements (Section 4.4). At the same time, I suggest to highlight that the validation was done against quite a suite of field measurements and variable pond conditions. This is a valuable aspect and could be stressed more. Many studies limit their validation to a single data set (e.g. one field experiment).

We think that most of the facts are performed in the main text. We added the names of expeditions once again to the conclusion.

Table 1 - I think that this is not needed.

The purpose of the table is clarifying for the reader which parameters are variable (and, consequently, are varied in the retrieval) and which are fixed in the model.

Table 2 - The pond code names seem to be an internal coding with almost no use for other studies. Using station names and dates as identifiers that link to field reports, Polarstern station numbers, and Pangaea data sets is suggested.

We put the station number in the case of Polarstern expedition.

- I suggest to re-arrange the columns and group retrieved/measured/difference (absolute, and %) for each: ice thickness and water depth. This eases evaluation of the performance.

Done.

- RMSD values could be given in units of e.g. 10ˆ-3 to save space and ease reading

Done.

---

## Author Comment (AC3) · 14 Feb 2018

Dear Editor and Editorial support, today is the deadline for submitting the answers to the reviewers to our manuscript. I am now uploading the answers to the reviewers. However, the revised version of the manuscript still requires a final editing. Therefore we are kindly asking for an extension to submit the revised manuscript according to your suggestions in https://editor.copernicus.org/index.php/tc-2017-150-EC1.pdf?_mdl=msover_md&_jrl=25&_lcm=oc108lcm109w&_acm=get_comm_file&_ms=60578&c=135714&salt=9427229 . It will greatly improve the manuscript to still do the final editing.

Please let us know your decision.

Best regards Georg Heygster

---

## Editor Decision (ED1)

**Referee #1 :**

You are absolutely right that just the optical depth, rather than the geometrical one, determines the reflectance. They are related by Eqs. (42). We consider all three parameters, $z$, $H$, and $\sigma_t$, as independent ones. We vary all of them independently when fitting spectra and don't make any additional assumptions about $\sigma_t$. (Except vertical variability). Of course, we don't have enough information to retrieve the vertical profile of $\sigma_t$, so we assume that we retrieve some constant effective value for a layer). Thus, all these three values are retrieved for every spectrum. In Table 2 we show only two of them just for comparison with the *in situ* measured values of $z$ and $H$. This information will also be added to the manuscript. However, nobody measures $\sigma_t$, so we don't show its values. But we added the retrieved values of $\sigma_t$ for the light and dark portions of the SHEBA pond (see the last paragraph of Sec. 4.3), where they are important for calculation of the scattering coefficient by bubbles.

Do I understand correctly (from your answers to the other referee) that you are best fitting the observed spectra with the model output spectra, based on triplets of H,z,σ$_t$ values? If that is the case,

    a) it should appear clearly in the abstract and in the methodology
    b) retrieved σ$_t$ should also appear in Table 2 (even if there is no measured equivalent) and something said about how these values compare to "sensible" values in the literature...

Also, captions for Figs 7, 8, 9, 10, 11 need to contain information about the general locations of each series.

We put in the text about the Polarstern cruise: "The stations, where ponds were observed, were located from 84°3N, 31°7E to 82°54N, 129°47E. For more information about the cruise, see Boetius et al. (2012) and Istomina et al. (2016)."
For Barrow and SHEBA the locations are given: Chukchi and Beaufort seas.

This is not enough. Captions should be self-explaining, with location referring to each pond shown

Fig 8 If these ponds were heterogeneous, then the exact location of the albedo measurement matters! Can this location be shown?

The exact point of the measurement can be seen in the photograph, where a person is taking observation from the light portion of the pond. Unfortunately, there is no photo for the dark one.

This is in Figure 7, not 8. I believe that what the referee would like to see, given the heterogeneity, and I agree, is where on the picture, the measurement was taken. If this info is not available, mention it in, the caption.

Referee #2:

data, which is hardly made by anyone for *in situ* measurements. As far as we're concerned, we think that all these points are stated in the Introduction. Also, according to your advice, we added these points to the Abstract.

Well, I believe this still remains cryptic for a non-specialist in the domain. I suggest that you add a short introductory paragraph at the beginning of section 2 (model description) with a tentative title such as: 2.1 New developments on melt pond reflections model. That paragraph would summarize your arguments presented here above.

> o How can or should this model be used in future (the outlook at the very end is rather unspecific and too general)?
> o What kind of scientific merit do the authors expect from this and following studies (applications of the model).

Of course, we cannot predict all possible merits. But some applications are obvious: such a model is absolutely necessary for satellite data processing in remote sensing of Arctic ice. Particularly, this model has served as a basis for the MPD (Melt Pond Detector) algorithm for melt pond fraction and sea ice albedo retrieval from MERIS data (Zege et al., 2015).

How did you change the conclusion accordingly?... This again is important for non-specialists to see the broader applications. From the detailed comments, nothing seems to have been changed in the conclusion.

> - The comparison with in-situ observations show differences of under-pond ice thickness and water depth of 50% and some even significantly higher. I do not follow the argumentation that this is satisfactory, in particular since there is very little discussion about this (see comments below). I consider these differences as more significant than the discussion reveals. In particular with respect to the under-pond (substrate) thickness, which should be the most important parameter to determine pond albedo.

Actually, the most important parameter that determines the pond albedo is the transport optical thickness of under-pond ice $\tau_t$ that is a product of the transport scattering coefficient $\sigma_t$ and ice thickness $H$: $\tau_t = \sigma_t H$. Partially this explains the retrieval error: $\tau_t$ is retrieved with much higher accuracy, however there is no way to compare it with a measured value. There could be also other different sources of error. First, the under-pond ice might not be flat, especially its lower boundary. In this case the optical retrieval gives some average value, while the *in situ* measurement gives a random value taken in some particular point. From this point of view the measurement makes a mistake, rather than the retrieval. The second source can be the presence of some impurities that affect the absorption spectrum. Additional absorption can affect the retrieval of the scattering coefficient and, consequently, of $H$. Besides, there could be other sources of uncertainties, like finite pond size, presence of snow in the receiver FOV, clouds in the sky etc. In view of that, the RMS error of 37% seems to us more than reasonable, especially given the fact that the microwave sounding methods fail absolutely in ice thickness retrieval, when ice is covered with a thin water layer.

a) I am sorry, but I don't see how you can judge "accuracy" if you don't know the real/measured value!?!..
b) I don't see these arguments developed clearly in the discussion. Where is this said precisely?

Abstract: The abstract may be significantly improved by adding more results and a statement that explicitly names the additional benefit and further applications of the model:
- Page1/Line15 (P1/L15): ... are examined: What is the result of the examination?

We added: "We find that atmospheric correction is necessary even for *in situ* measurements. Thus, an atmospheric correction procedure has been used in the model verification"

This is not enough, in my view...compared to what the reviewer asked

- P1/L17: "good performance" this is rather relative, good in what measure?

How can we measure the adequacy of a model or a theory? This is rather quality, than quantity measure.

Well, to me, by seeing how well it reproduces the observations… and you have the quantitative answer to that in your data…

To find the best fit solution we use the multidimensional Newton-Raphson method with the singular value decomposition of the pseudo-inverse matrix. We really think that the discussion of the method lies far beyond the paper scope, but the method name is added to the manuscript. Adding computational details will make the understanding of the work only harder. Also we are sure that the particular method of searching solution doesn't matter for model verification. It is enough that we find such values of the three pond parameters that give the best fit of spectra in the sense the least squares.

o How did the authors derive that these are the three main parameters. What other parameters were analyzed?

See above our answer about the role of these three parameters. Additionally we can note that refractive indices and absorption spectra of ice and water were not analyzed, because they are fixed, and sediment concentration was not analyzed, because we have no information about polluting substances. So, no more parameters can be analyzed from the point of view of albedo spectrum. Another question is that the transport scattering coefficient consists of the contributions of air bubbles and brine inclusions and thus is determined by their concentrations. Their relationships are considered in detail in Sec. 2.2c and 4.3.

o What about the transport coefficients? How were they studied/discussed?
o How are the thicknesses retrieved?

All three parameters are retrieved in the same manner. They comprise a 3d-vector, which is varied to provide the best fit of spectra. We added this phrase to Sec. 4.1.

a) I did not find the Newton-Raphson method mentioned in the paper

b) Retrieval of the parameters… I think this is something that was unclear to all readers… it is however a crucial output of the paper!... I think it would deserve a "method supplementary material" in which you show on an example how the three parameters are retrieved. Surely there must be a (graphic?) way to show that you chose the best combination of the three parameters by reducing RMS. How do we know this is a "univoqual" solution in each case? How do we know there are no multiple combinations of the three variables resulting in a similarly good fitting of the spectra?

> - Section 4.4 should be the main discussion of the comparisons. This is too short and somewhat superficial.
> o Where do these rather large differences of 50% come from? I do see various reasons in e.g. pond depth distributions, non-planar interfaces, footprint of sensors compared to pond properties. But this needs to be discussed in more detail.
> o What precision may/can be expected in such models?

> o What determines the uncertainties? Which of the given assumptions might not be ideal, but what would it mean to adapt this? It is most likely not realistic within this study, but some additional discussion would be useful and interesting for further studies.

Throughout the manuscript, making the derivations, we stated the assumptions we use in the model. Surely, every assumption is some approximation or idealization and any of them can limit applicability and accuracy of results. However, the perfect fit of the measured and modeled spectra is a proof that these assumptions were reasonable.

> - With respect to those differences: As discussed, impurities are mostly low in the ponds,
> so the result is mostly based on scattering (not absorption). In this case, the retrieved spectral shape may be expected to be in good agreement, while amplitude is the main aspect of evaluation. But if then the simulated differences are still around 50% for the under-ice thickness this is somewhat surprising to me. I agree that the RMSE match is quite good if not excellent, but may be not because of the right thicknesses, but other parameters in the model. This should be discussed more.

We think this question is answered in the section *'General comments'* (the 4th question). (Also note that the mean error for ice thickness is 37%, not 50%).

Here the reviewer makes explicit suggestions on the type of questions that should be answered in this discussion section of the text…not only as a response to him!...I don't think this has been done. It is not enough to answer that these matters have been "stated throughout the manuscript".

For all the comments on the Conclusion, the same applies… it is not enough to answer to the reviewer and not change things in the text, and state that "most of the facts are performed in the

main text". Please develop the conclusion as a summary of the main outputs and perspectives of your work.

---

## Author Response (AR2)

Dear Editor,

First of all, we'd like to apologize for some misunderstanding. One of our co-authors, trying to improve our English, made a lot of corrections throughout the manuscript. Unfortunately, due to some Microsoft Word bug, only his changes remained highlighted, so that you could get the impression that no more changes (according to referees' notes) were made. Whereas a lot of corrections were made according to referees' comments, and only few of them were just answered without changes in the manuscript. Sometimes we somewhat disagreed with the Reviewer's opinion. In these cases we tried to explain our position in detail in our answers to reviewers.

We are grateful to you for your comments. This time, to avoid misunderstanding, we highlighted only the changes made according to your comments. Please, find our answers point-by-point below.

Do I understand correctly (from your answers to the other referee) that you are best fitting the observed spectra with the model output spectra, based on triplets of $H, z, \sigma_t$ values?

Yes, you are absolutely right.

If that is the case,
a) it should appear clearly in the abstract and in the methodology

Added: page 1, lines 20-21 and 25-26.

Also see: p. 14, l. 16-19.

b) retrieved $\sigma_t$ should also appear in Table 2 (even if there is no measured equivalent) and something said about how these values compare to "sensible" values in the literature…

The retrieved $\sigma_t$ added to Table 2.

A paragraph about the range of its values is added in p. 8, l.23 - p.9, l.5 and about the retrieval p.16, l.4-6

This is not enough. Captions should be self-explaining, with location referring to each pond shown

Locations are added to captions of Figs. 6-10.

I believe that what the referee would like to see, given the heterogeneity, and I agree, is where on the picture, the measurement was taken. If this info is not available, mention it in, the caption.

We added to the caption: "The left pond is heterogeneous. The sensor was placed approximately in the center of the photograph, about 1m from the pond edge."

Well, I believe this still remains cryptic for a non-specialist in the domain. I suggest that you add a short introductory paragraph at the beginning of section 2 (model description) with a tentative title such as: 2.1 New developments on melt pond reflections model. That paragraph would summarize your arguments presented here above.

We thought that derivation of formulas in Sec.2.1 would help a reader in understanding the approach. However, now we see, by yours and reviewer's reaction, that, vice versa, it was only confusing. We completely rewrote Section 2.1 (NOTE that it is NOT highlighted, because it is completely new). Now there cannot be any misunderstanding about what is new and what are the results of other authors.

How did you change the conclusion accordingly?... This again is important for non-specialists to see the broader applications. From the detailed comments, nothing seems to have been changed in the conclusion.

Actually we had somewhat corrected the conclusion regarding reviewers' recommendations. Now we added another one paragraph to the conclusion about possible applications.

Actually, the most important parameter that determines the pond albedo is the transport optical thickness of under-pond ice $\tau_t$ that is a product of the transport scattering coefficient $\sigma_t$ and ice thickness $H$: $\tau_t = \sigma_t H$. Partially this explains the retrieval error: $\tau_t$ is retrieved with much higher accuracy, however there is no way to compare it with a measured value. There could be also other different sources of error. First, the under-pond ice might not be flat, especially its lower boundary. In this case the optical retrieval gives some average value, while the *in situ* measurement gives a random value taken in some particular point. From this point of view the measurement makes a mistake, rather than the retrieval. The second source can be the presence of some impurities that affect the absorption spectrum. Additional absorption can affect the retrieval of the scattering coefficient and, consequently, of $H$. Besides, there could be other sources of uncertainties, like finite pond size, presence of snow in the receiver FOV, clouds in the sky etc. In view of that, the RMS error of 37% seems to us more than reasonable, especially given the fact that the microwave sounding methods fail absolutely in ice thickness retrieval, when ice is covered with a thin water layer.

a) I am sorry, but I don't see how you can judge "accuracy" if you don't know the real/measured value!?!..
We meant here the simulation with synthetic data. We can judge about the retrieval accuracy in the closed numeric experiment.

b) I don't see these arguments developed clearly in the discussion. Where is this said precisely?
We added these arguments in p.15, l. 17-27.

Abstract: The abstract may be significantly improved by adding more results and a statement that explicitly names the additional benefit and further applications of the model:
- Page1/Line15 (P1/L15): ... are examined: What is the result of the examination?

We added: "We find that atmospheric correction is necessary even for *in situ* measurements. Thus, an atmospheric correction procedure has been used in the model verification"

This is not enough, in my view...compared to what the reviewer asked
The phrase "...that atmospheric correction is necessary..." was the answer to the question "What is the result of the examination?"
What concerns 'adding more results,' the main result of the work is that the reflective properties of a melt pond are determined by three parameters, z,H,$\sigma_t$ (and how). We put this statement in p.1, l. 20-21.

Well, to me, by seeing how well it reproduces the observations... and you have the quantitative answer to that in your data...
All the RMSD both for spectra and for retrieved values are given (Table 2 and Section 4.4). The correlation coefficient $R^2$ of the retrieved and *in situ* measured values is also given (Section 4.4 and Figure 13). We also added a phrase about the maximal error to the Abstract (p. 1, l. 27).

a) I did not find the Newton-Raphson method mentioned in the paper
We put it in p. 13, l. 6.

b) Retrieval of the parameters... I think this is something that was unclear to all readers... it is however a crucial output of the paper!... I think it would deserve a "method supplementary material" in which you show on an example how the three parameters are retrieved. Surely there must be a (graphic?) way to show that you chose the best combination of the three parameters by reducing RMS. How do we know this is a "univoqual" solution in each case? How do we know there are no multiple combinations of the three variables resulting in a similarly good fitting of the spectra?

This is the question we do want to avoid in this paper. The main point in this paper is that the three-parametric model works. There exist such values of z, H, σt that provide a good coincidence of the measured and modeled spectra. Any reader can take the values from the Table, put them into the formulas, get the spectra shown in plots, and make sure that they fit the measured spectra given in the supplement. We do not propose and discuss the method for retrieving these values for the measured spectra. We only show that the three-parametric model is able to reproduce the albedo spectrum in the visible range with RMSD that does not exceed 1.5% for a wide variety of melt pond.

Retrieval of medium optical parameters from measured spectra is an inverse ill-posed problem, which is a separate wide section of the contemporary mathematical science. E.g., we cannot be sure that there is no another combination that provides better fitting, because the RMSD function may have several local minima. Moreover, these local minima (and even some maxima) may lie in the range that gives the variety of spectra, all inside the experimental error. In which case there is no way to choose the 'best' solution and *a priori* information is needed for regularization.

This problem lies far outside our humble work and requires further investigations. Any effort to consider it in this particular paper, we are sure, can only confuse a reader and make reading much more difficult. Nevertheless, for those who are interested, we put a reference to the description of the method we use (p. 13, l.7).

Here the reviewer makes explicit suggestions on the type of questions that should be answered in this discussion section of the text…not only as a response to him!...I don't think this has been done. It is not enough to answer that these matters have been "stated throughout the manuscript".
We believe that the comprehensive answers are given in p. 1, l. 25-27 and p.15, l. 17-27.

For all the comments on the Conclusion, the same applies… it is not enough to answer to the reviewer and not change things in the text, and state that "most of the facts are performed in the main text". Please develop the conclusion as a summary of the main outputs and perspectives of your work.
Conclusion is edited strongly as a whole. In particular, some sentences added in p.16, l. 11-13, 17-19, 24-25, 29-31, and p.17, l. 7-14, 17-19.
Now it is not a brief but very detailed summary of our work and its possible perspectives.

[revised manuscript text omitted]

---

## Editor Decision (ED2)

Dear Authors,

Thank you very much for your updated review. I think we are very close to acceptance. I can see two last items I would like to see fulfilled:

**a)**

*Well, I believe this still remains cryptic for a non-specialist in the domain. I suggest that you add a short introductory paragraph at the beginning of section 2 (model description) with a tentative title such as: 2.1 New developments on melt pond reflections model. That paragraph would summarize your arguments presented here above.*
*We thought that derivation of formulas in Sec.2.1 would help a reader in understanding the approach.*
*However, now we see, by yours and reviewer's reaction, that, vice versa, it was only confusing. We completely rewrote Section 2.1 (NOTE that it is NOT highlighted, because it is completely new). Now there cannot be any misunderstanding about what is new and what are the results of other authors.*

Sorry, but I still think it is still not clear what is the original input of this paper modelwise in this chapter 2...it remains "diluted" in the several pages descriptions of the model. **PLEASE**, at least clearly state the originality in a few sentences either before section 2.1 or as a conclusive paragraph at the end of section 2.

**b)**

I have made a few "formal" suggestions for changes in the phrasing of the newly added sections, as follows:

Page 15:
-**4.4 Verification results**
The retrieved and measured pond parameters (melt water depth, and underlying ice thickness, and transport scattering
15 coefficient), as..

Page 15:
-25 uncertainties, like finite pond size, presence of snow in the receiver field-of-view, clouds in the sky aso. Considering this, the retrieval of the underlying ice thickness seems to be rather reasonable. Let us note the fact that microwave sounding methods completely fail absolutely in ice thickness retrieval, when ice is covered with a water layer of indefinite thickness.

Page 16:
-more pond parameters will not help improving the model and make it closer to reality. We would attribute potentially appearing remaining differences between observed and modeled spectra to possible the potential impact of sediments within the ice.
What about the other potential parameters explaining discrepancy?...under pond ic not flat, presence of impurities, finite pond size,snow in receiver FOV, clouds..?..Why only selecting sediments?

Page 17:

-10 In  turn, the amount of melt ponds on Arctic sea ice determines the sea ice reflectance and transmittance and thus allows estimating the energy balance above, within, and under sea ice and its response to climate change. The temporal evolution of melt ponds consists of melt stages, which are specific to sea ice type (landfast ice, first-, second-, or multi-year ice). The spring melt pond fraction predicts the autumn Arctic sea-ice  extent (?). Therefore, the melt pond fraction dataset obtained from satellite data is required to derive the sea ice  extent (?) and type during summer melt.

---

## Author Response (AR3)

Dear Editor,

First of all, let us say that we are grateful for your expert help. Our answers to your last comments are below.

In addition to the revised version of the manuscript, we are submitting as a supplement a version of the manuscript with all changes relative to the last version highlighted.

> Dear Authors,
> Thank you very much for your updated review. I think we are very close to acceptance. I can see two last items I would like to see fulfilled:
> **a)**
> Sorry, but I still think it is still not clear what is the original input of this paper modelwise in this chapter 2…it remains "diluted" in the several pages descriptions of the model. **PLEASE**, at least clearly state the originality in a few sentences either before section 2.1 or as a conclusive paragraph at the end of section 2.

We put additional explanations in p. 4, l. 9-13 and p. 5, l. 6-12. Also the originality was stated in the introduction (p. 2, l. 19-28).

> **b)**
> I have made a few "formal" suggestions for changes in the phrasing of the newly added sections, as follows:
>
> Page 15:

Corrected.

> Page 16:
> What about the other potential parameters explaining discrepancy?...under pond ic not flat, presence of impurities, finite pond size,snow in receiver FOV, clouds..?..Why only selecting sediments?

You are right; there can be external factors that affect the spectrum. We deleted these two last sentences to avoid the excessive discussions.

> Page 17: -10

Corrected.

Thank you for suggestions for changes in the phrasing of the newly added sections.